# Fed-ARPL: Adaptive and Reciprocal Prototype Learning for Semi-supervised Federated Learning

## Abstract

Federated Semi-supervised Learning (FSSL) enables collaborative training by leveraging a small labeled dataset on a central server and vast unlabeled data across clients. However, existing frameworks are hampered by two challenges: an initial Cold-start phase, where strict pseudo-label filtering criteria impede the use of unlabeled data, and a subsequent Knowledge Bottleneck, where the model's performance is capped by the server's limited and potentially biased labeled data. To address these challenges, we propose Fed-ARPL, a novel Adaptive and Reciprocal Prototype Learning framework that implements a meticulously designed three-phase learning strategy. First, a Warm-up Phase employs an adaptive thresholding mechanism to resolve the Cold-start dilemma, dynamically adjusting the pseudo-label confidence to accelerate initial convergence and establish a stable feature space. Next, a Teacher-Guided Phase leverages the server's reliable prototypes to provide unified, one-way guidance, steering all clients toward a consistent and well-structured representation. Finally, to break the Knowledge Bottleneck, the framework culminates in a Student-Feedback Phase, establishing a reciprocal paradigm where high-performing clients contribute their refined local prototypes to enrich the global consensus. Comprehensive experiments validate the effectiveness of our Fed-ARPL framework, showcasing its state-of-the-art (SOTA) performance on several widely-recognized benchmark datasets.

## 1 Introduction

Federated Learning (FL) McMahan et al. (2017) is a distributed training paradigm that enables collaborative model development across clients without sharing their private data, and extensive research has been conducted across various aspects of the field Yuan & Li (2022); Li et al. (2021); Yazdinejad et al. (2024). A primary challenge in practical FL, however, is the scarcity of labeled data. Clients often lack the motivation to label data, and even when they do, the annotations may not always be reliable. To address this limitation, Federated Semi-supervised Learning (FSSL) Diao et al. (2022) has emerged as a critical research direction. FSSL operates under a pragmatic setting where a central server holds a small, labeled dataset to guide the learning process across numerous clients that possess vast amounts of unlabeled data. This configuration is representative of many real-world applications, such as in healthcare Kassem et al. (2022); Ma et al. (2024) and transportation Zhu et al. (2021), where it is imperative to train powerful global models using only a small fraction of labeled data.

Despite its promise, current research in FSSL is constrained by the high and static thresholds provided by the server in the initial phase Diao et al. (2022); Jeong et al. (2020). This reliance leads to a severe *'Cold-start'* problem: the nascent global model is too weak to generate pseudo-labels that meet this stringent criterion, trapping it in a cycle where client data remains dormant and cannot be effectively used, resulting in poor data utilization efficiency in the initial stages. Furthermore, these frameworks are afflicted by a limitation during training: the *'Knowledge Bottleneck'*. Here, the global *'teacher'* model's knowledge is fundamentally confined by the server's small labeled dataset. This forces clients to merely mimic a teacher with a constrained worldview, preventing the system from harnessing the federation's collective intelligence to discover a truly superior global optimum.

To alleviate the aforementioned challenges, we reconceptualize the FSSL process not as a static aggregation mechanism, but as a dynamic, evolving pedagogical journey. Our key insight is that these two challenges must be solved sequentially and synergistically. To dismantle the Cold-start deadlock, we first introduce a co-designed dual strategy: adaptive thresholding empowers global model to dynamically adjust its expectations, activating client data from the outset. Simultaneously, to counteract the risk of noise from early, low-quality pseudo-labels, we introduce server-guided prototypical learning Snell et al. (2017), where server-computed prototypes serve as authoritative learning templates to regularize and unify client training. Furthermore, to shatter the Knowledge Bottleneck, we evolve the role of prototypes from this top-down instructional tool to a vehicle for bottom-up student feedback. This establishes a reciprocal learning loop where high-achieving clients contribute their refined local prototypes to enrich the global consensus, allowing the system to transcend the teacher's inherent limitations and break the performance ceiling.

To bring this pedagogical philosophy into practice, we introduce Fed-ARPL, a novel Adaptive and Reciprocal Prototypical Learning framework. Fed-ARPL employs a meticulously designed three-phase training strategy, where each subsequent phase is intelligently activated to address a new layer of challenges. The process commences with the Warm-up Phase, which exclusively employs our adaptive thresholding mechanism to break the initial learning inertia. By dynamically lowering the bar for pseudo-label acceptance, this phase rapidly activates client participation, ensuring that a stable and well-structured feature space is formed as a crucial foundation. Once this foundation is mature, the framework seamlessly transitions into the Teacher-Guided Phase. Here, reliable class prototypes from the server are introduced as knowledge anchors, providing unified, one-way guidance to prevent clients from diverging and to steer the entire federation efficiently toward a common objective. Finally, to shatter the Knowledge Bottleneck, the framework transitions into the Student-Feedback Phase. This stage unlocks the full potential of reciprocal learning by additionally empowering high-performing clients to contribute their own refined local prototypes. These insights are then intelligently synthesized with the teacher's knowledge at the server, creating a synergistic global consensus that transcends the limitations of any single agent and breaks the performance ceiling.

Our main contributions are summarized as follows:

- We introduce Fed-ARPL, a novel and systematic framework for FSSL that, to our knowledge, is the first to conceptualize the learning process as a dynamic, evolving pedagogical relationship.
- We design an adaptive thresholding mechanism that effectively addresses the Cold-start predicament in FSSL, thereby significantly accelerating the model's early-stage convergence.
- We propose an innovative three-phase prototypical learning strategy that evolves from unified server-guided learning to a final reciprocal student-feedback mechanism, a dual approach that successfully breaks the Knowledge Bottleneck.
- Comprehensive experiments demonstrate that Fed-ARPL achieves state-of-the-art (SOTA) performance on multiple benchmark datasets, with ablation studies validating the effectiveness of each proposed component.

## 2 RELATED WORKS

### 2.1 FEDERATED SEMI-SUPERVISED LEARNING

Federated Semi-Supervised Learning adapts techniques from centralized Semi-Supervised Learning, which primarily revolves around the core ideas of pseudo-labeling Lee et al. (2013) and consistency regularization Sajjadi et al. (2016) . The seminal work FixMatch Sohn et al. (2020) integrated these ideas, establishing a powerful baseline by generating pseudo-labels on weakly-augmented data and enforcing consistency with predictions on strongly-augmented counterparts.

FSSL methods focus on adapting this powerful pseudo-labeling and consistency paradigm to the unique constraints of the federated context. FedMatch Jeong et al. (2020) introduced an inter-client consistency loss to regularize local training across different clients. SemiFL Diao et al. (2022) proposed an alternating training scheme, which decouples server fine-tuning on labeled data from client training on pseudo-labels, improving overall stability and performance. To further refine the

quality of pseudo-labels, subsequent works have explored more advanced techniques. For instance, $(FL)^2$ Lee et al. (2024) tackled the confirmation bias issue with sharpness-aware regularization and introduced client-specific adaptive thresholding. Similarly, BSemiFL Wang et al. (2025) and the recent FedLGMatch Zhao et al. (2025) proposed leveraging both global and local model predictions for more robust re-labeling, attempting to mitigate noise in Non-IID settings. These works, alongside others employing techniques like knowledge distillation Liu et al. (2021) and advanced aggregation rules Zhang et al. (2021), have significantly elevated the sophistication of the learning process.

Despite these advancements, the aforementioned **Cold-Start** problem remains a largely unaddressed issue that severely impairs learning efficiency. The reliance on a static, high-confidence threshold means that in the early training phase, the nascent global model is too weak to generate useful pseudo-labels. This leaves the vast unlabeled data on clients dormant, causing significant training inertia. Our work addresses this specific challenge by introducing a server-driven **Adaptive Thresholding** mechanism. This mechanism dynamically adjusts the confidence requirement based on the evolving proficiency of the global teacher model, thereby activating client data from the outset and effectively breaking the initial learning impasse.

## 2.2 PROTOTYPICAL LEARNING IN FEDERATED LEARNING

Prototypical learning, originating from Prototypical Networks Snell et al. (2017), provides an effective method for representation learning by classifying samples based on their distance to class prototypes. In federated learning, its primary application has been to mitigate the effects of statistical heterogeneity. Representative works like FedProto Tan et al. (2022) leverage this concept by having clients communicate low-dimensional class prototypes for server-side aggregation. The resulting global prototypes are then broadcast back to regularize local client training, ensuring local models do not diverge excessively from the global consensus. This paradigm of using prototypes for top-down regularization is a common thread in subsequent research Tran et al. (2024).

While these methods effectively use prototypes for regularization, the presence of a labeled dataset at the server fundamentally alters the potential of prototypical learning. It enables a transition from a peer-to-peer knowledge alignment to the structured pedagogical process that our work is designed to exploit. In this context, our Fed-ARPL framework redefines the role of prototypes by introducing a dynamic role evolution. We first employ them in a One-Way Prototypical Guidance phase, where server-computed prototypes serve as unambiguous "knowledge anchors" to stabilize the early, noise-prone training phase. Crucially, this is merely a preparatory step. Our key innovation lies in the subsequent phase, where prototypes become the vehicle for reciprocal co-creation. By enabling proficient clients to contribute their high-quality local prototypes, we facilitate a "student-feedback" loop that enriches the global consensus, allowing the system to break the very "Knowledge Bottleneck" that prior methods leave unaddressed.

## 3 METHODOLOGY

To address the challenges of cold-start inertia and the knowledge bottleneck, we propose **Fed-ARPL**, a novel FSSL framework that orchestrates the learning process through an adaptive, three-phase strategy. We first formalize the FSSL setting and outline the baseline workflow upon which our method is built, before detailing the core components of Fed-ARPL.

## 3.1 PROBLEM SETTING

We consider the "labels-at-server" FSSL setting. The system consists of a central server holding a small labeled dataset $\mathcal{D}_s$ and $M$ clients, each with a local unlabeled dataset $\mathcal{D}_k^u$. Our framework is built upon the alternating training procedure established in prior works Diao et al. (2022). A typical communication round $t$ in this baseline workflow involves three key steps. First, the server trains its model on $\mathcal{D}_s$ and distributes the resulting global model $\theta_g^t$ to a subset of clients. Second, each selected client $C_k$ uses the received $\theta_g^t$ to generate pseudo-labels for its local data. Following the FixMatch paradigm Sohn et al. (2020), the client then updates its local model $\theta_k^t$ by minimizing a

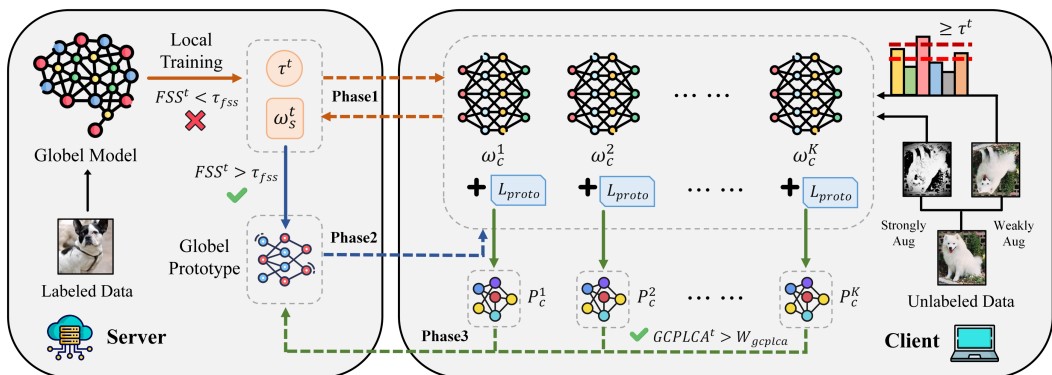

Figure 1: The overall architecture of Fed-ARPL, which uses a three-phase strategy to address the Cold-start and Knowledge Bottleneck challenges.

consistency regularization loss. For a mini-batch $\mathcal{B}_k^u$, this objective is formulated as:

$$\mathcal{L}_k^u = \frac{1}{|\mathcal{B}_k^u|} \sum_{x \in \mathcal{B}_k^u} \mathbb{I}(\max(q_g) > \tau) \cdot \mathcal{H}(\hat{y}_g, q_k). \tag{1}$$

Here, $q_g = p_{\theta_g^t}(y|\omega(x))$ and $q_k = p_{\theta_k^t}(y|\Omega(x))$ are predictions on weakly and strongly augmented views, respectively, $\hat{y}_g$ is the pseudo-label derived from $q_g$, and $\tau$ is a fixed high-confidence threshold. Finally, the clients upload their updated models, which the server aggregates using Federated Averaging McMahan et al. (2017) to initialize its model for the next round.

The core of Fed-ARPL lies in its dynamic adjustment of the client-side learning objective, $\mathcal{L}_k$, and the intelligent evolution of the global prototypes, $\mathcal{P}_g$. The transition between phases is governed by automated, data-driven criteria, ensuring that each mechanism is deployed at the opportune moment. An overview of the entire Fed-ARPL workflow is depicted in Figure 1.

### 3.2 PHASE 1: WARM-UP VIA ADAPTIVE THRESHOLDING

**Objective.** The primary goal of the initial phase is to rapidly overcome the cold-start problem by maximizing the utilization of unlabeled data from the very first round.

**Client-Side Training.** In this phase, each client $C_k \in \mathcal{C}_{sel}^t$ trains its local model by minimizing a standard semi-supervised consistency loss. Our key intervention in this phase is to replace the static threshold $\tau$ in the baseline client objective (Eq. 1) with a dynamic, server-provided adaptive threshold $\tau^t$.

**Server-Side Adaptive Threshold Generation.** The server computes the global adaptive threshold $\tau^t$ based on its own model's confidence on the clean, labeled dataset $\mathcal{D}_s$. This provides a unified and stable signal of the teacher's current proficiency to all clients. It is calculated as the mean of the maximum softmax probabilities (MSP) over $\mathcal{D}_s$:

$$\tau^t = \mathbb{E}_{(x,y) \in \mathcal{D}_s}[\max(p_{\theta_g^t}(y|x))]. \tag{2}$$

**Transition Criterion.** The framework transitions to Phase 2 when the feature space becomes sufficiently structured. We quantify this using the Feature-Space Separation (FSS) score (Eq. 3). Grounded in the principles of metric learning Snell et al. (2017); Schroff et al. (2015), which aim to maximize inter-class separation while minimizing intra-class variance, a high and stable FSS score serves as a reliable indicator that the model is ready for prototype-based guidance. The transition is triggered when FSS meets predefined stability and magnitude criteria over a window $W_{fss}$.

$$\text{FSS}^t = \frac{1}{K} \sum_{c=1}^{K} \frac{\min_{j \neq c} \|\mathbf{p}_c^s - \mathbf{p}_j^s\|_2}{\mathbb{E}_{x_i:y_i=c}\|h_\phi(x_i) - \mathbf{p}_c^s\|_2 + \epsilon} \tag{3}$$

### 3.3 Phase 2: Teacher-Guided Prototypical Learning

**Objective.** Upon entering Phase 2, the framework's objective shifts to accelerating convergence and enforcing a globally consistent feature manifold by leveraging the server's reliable knowledge.

**Server-Side Prototype Guidance.** The server, acting as the teacher, computes its prototypes $\mathcal{P}_s^t = \{\mathbf{p}_c^s\}$ from $\mathcal{D}_s$. These are designated as the global prototypes for the round, $\mathcal{P}_g^t \leftarrow \mathcal{P}_s^t$, and are disseminated to the clients along with the global model $\theta_g^t$ and the adaptive threshold $\tau^t$.

**Client-Side Guided Training.** The client's local learning objective is now augmented with a prototype consistency loss, $\mathcal{L}_{proto}$, to enforce alignment with the global guidance. The total loss for client $C_k$ becomes:

$$\mathcal{L}_k^{(2)} = \mathcal{L}_k^{(1)} + \lambda_{plc} \cdot \mathcal{L}_{proto}, \tag{4}$$

where $\mathcal{L}_k^{(1)}$ is the adaptive-threshold consistency loss from Eq. equation 1, and $\lambda_{plc}$ is a hyperparameter balancing the two loss terms. The prototype loss is formulated as the mean squared error between the feature embeddings of pseudo-labeled samples and their corresponding global prototypes:

$$\mathcal{L}_{proto} = \frac{1}{|\mathcal{B}'|} \sum_{x_j^k \in \mathcal{B}'} \|h_{\phi_k^t}(x_j^k) - \mathbf{p}_{\hat{y}_j^g}^g\|_2^2, \tag{5}$$

where $\mathcal{B}' \subseteq \mathcal{B}_k^u$ is the subset of the mini-batch with pseudo-labels that satisfy the threshold $\tau^t$.

**Transition Criterion.** The transition to the final, reciprocal phase is predicated on the clients reaching a sufficient level of proficiency. This is measured by the Global Client Pseudo-Label Confidence Aggregation (GCPLCA) score, which aggregates the local average confidence, $\mathbb{E}_{x \in \mathcal{D}_k^u}[\max(p_{\theta_g^t}(y|x))]$, from each client. The GCPLCA score serves as a proxy for the system's collective *epistemic certainty*, a concept where high confidence is indicative of a converged and reliable model state Guo et al. (2017); Sohn et al. (2020). A high and stable GCPLCA thus signals that clients are ready to contribute reliable feedback. The transition occurs when GCPLCA meets predefined stability and magnitude criteria over a window $W_{gcplca}$.

### 3.4 Phase 3: Reciprocal Prototypical Learning

**Objective.** The final phase is designed to break the knowledge bottleneck by establishing a reciprocal learning loop, where clients' insights are used to enrich the global consensus.

**Client-Side Feedback Generation.** Each client $C_k$ continues to train using the same objective as in Phase 2, i.e., $\mathcal{L}_k^{(3)} = \mathcal{L}_k^{(2)}$. Concurrently, it acts as a knowledge contributor. It identifies a high-confidence subset of its data, $\mathcal{D}_k^{u'} = \{x \in \mathcal{D}_k^u \mid \max(p_{\theta_g^t}(y|x)) > \tau'\}$, using a stricter threshold $\tau' > \tau^t$. This rigorous filtering ensures that prototypes are constructed exclusively from high-quality representations, minimizing noise from ambiguous data. From this reliable subset, it computes its local prototypes $\mathcal{P}_k^t = \{\mathbf{p}_c^k\}$, where each prototype is the mean feature embedding over samples with pseudo-label $c$, calculated as $\mathbf{p}_c^k = (1/|\mathcal{D}_{k,c}^{u'}|) \sum_{x \in \mathcal{D}_{k,c}^{u'}} h_{\phi_k^t}(x)$. These prototypes are then uploaded to the server along with the client's model parameters.

**Server-Side Reciprocal Update.** Following the standard aggregation of the parameters of the client model McMahan et al. (2017), the server performs a reciprocal update of the global prototypes. It first aggregates the received client prototypes $\{\mathbf{p}_c^k\}$ to form the student consensus prototypes, $\bar{\mathcal{P}}^t = \{\bar{\mathbf{p}}_c^t\}$. Specifically, the consensus prototype for each class $c$ is computed by averaging the client prototypes, weighted by the number of high-confidence samples $n_c^k$ used for their computation on each client $k \in \mathcal{C}_{sel}^t$:

$$\bar{\mathbf{p}}_c^t = \frac{\sum_{k \in \mathcal{C}_{sel}^t} n_c^k \cdot \mathbf{p}_c^k}{\sum_{k \in \mathcal{C}_{sel}^t} n_c^k}. \tag{6}$$

This weighting strategy naturally diminishes the influence of clients with sparse or unreliable data for specific classes, mitigating the impact of statistical heterogeneity. It then combines its own

Table 1: Test accuracy (%) comparison of FSSL methods on CIFAR-10, CIFAR-100, and SVHN under different data distributions. $N_s$ denotes the number of labeled samples at the server. Results are reported as mean $\pm$ standard deviation over 3 random seeds. The best performance in each column is highlighted in bold.

| Data Distribution | Method | CIFAR-100 | | CIFAR-10 | | SVHN | |
|---|---|---|---|---|---|---|---|
| | | $N_s = 2500$ | $N_s = 10000$ | $N_s = 250$ | $N_s = 4000$ | $N_s = 250$ | $N_s = 1000$ |
| Non-IID (Dir $\alpha = 0.2$) | SemiFL | $43.32_{\pm 0.15}$ | $60.29_{\pm 0.30}$ | $67.59_{\pm 0.45}$ | $87.66_{\pm 0.33}$ | $91.55_{\pm 0.25}$ | $91.45_{\pm 0.28}$ |
| | FedMatch | $21.21_{\pm 2.95}$ | $29.85_{\pm 2.80}$ | $44.35_{\pm 2.50}$ | $58.10_{\pm 1.80}$ | $68.55_{\pm 2.90}$ | $70.13_{\pm 2.50}$ |
| | FL$^2$ | $31.95_{\pm 1.00}$ | $44.23_{\pm 1.50}$ | $60.69_{\pm 1.00}$ | $76.66_{\pm 1.20}$ | $84.39_{\pm 1.10}$ | $83.87_{\pm 1.30}$ |
| | pFedKnow | $19.10_{\pm 2.10}$ | $31.50_{\pm 2.50}$ | $52.72_{\pm 1.90}$ | $64.33_{\pm 1.70}$ | $58.81_{\pm 2.80}$ | $62.92_{\pm 2.60}$ |
| | **Fed-ARPL** | $\mathbf{47.43}_{\pm 0.37}$ | $\mathbf{61.84}_{\pm 0.20}$ | $\mathbf{69.11}_{\pm 0.52}$ | $\mathbf{87.96}_{\pm 0.25}$ | $\mathbf{92.42}_{\pm 0.21}$ | $\mathbf{92.12}_{\pm 0.22}$ |
| Non-IID (Dir $\alpha = 0.5$) | SemiFL | $43.53_{\pm 0.13}$ | $61.17_{\pm 0.18}$ | $73.66_{\pm 0.21}$ | $90.73_{\pm 0.20}$ | $93.75_{\pm 0.20}$ | $92.04_{\pm 0.25}$ |
| | FedMatch | $22.33_{\pm 2.90}$ | $30.41_{\pm 2.70}$ | $45.18_{\pm 1.60}$ | $59.35_{\pm 1.50}$ | $69.80_{\pm 2.60}$ | $71.25_{\pm 2.20}$ |
| | FL$^2$ | $32.81_{\pm 1.20}$ | $45.42_{\pm 1.30}$ | $62.21_{\pm 0.90}$ | $78.03_{\pm 1.10}$ | $85.92_{\pm 1.00}$ | $84.39_{\pm 1.20}$ |
| | pFedKnow | $19.65_{\pm 2.20}$ | $32.86_{\pm 2.40}$ | $54.15_{\pm 1.70}$ | $65.89_{\pm 1.50}$ | $60.06_{\pm 2.70}$ | $70.11_{\pm 2.40}$ |
| | **Fed-ARPL** | $\mathbf{50.97}_{\pm 0.22}$ | $\mathbf{62.44}_{\pm 0.20}$ | $\mathbf{75.13}_{\pm 0.31}$ | $\mathbf{90.98}_{\pm 0.15}$ | $\mathbf{94.17}_{\pm 0.18}$ | $\mathbf{92.53}_{\pm 0.20}$ |
| Non-IID (Dir $\alpha = 0.8$) | SemiFL | $46.46_{\pm 0.10}$ | $61.94_{\pm 0.23}$ | $\mathbf{78.37}_{\pm 0.30}$ | $91.43_{\pm 0.22}$ | $93.88_{\pm 0.22}$ | $92.69_{\pm 0.23}$ |
| | FedMatch | $22.51_{\pm 3.20}$ | $31.02_{\pm 2.50}$ | $45.33_{\pm 1.20}$ | $59.81_{\pm 1.20}$ | $70.52_{\pm 2.50}$ | $71.99_{\pm 2.10}$ |
| | FL$^2$ | $33.54_{\pm 1.40}$ | $45.51_{\pm 1.20}$ | $60.84_{\pm 1.20}$ | $75.98_{\pm 1.40}$ | $88.65_{\pm 1.10}$ | $84.91_{\pm 1.20}$ |
| | pFedKnow | $20.03_{\pm 1.80}$ | $33.15_{\pm 2.20}$ | $54.57_{\pm 1.50}$ | $66.41_{\pm 1.40}$ | $61.38_{\pm 2.60}$ | $60.83_{\pm 2.30}$ |
| | **Fed-ARPL** | $\mathbf{51.76}_{\pm 0.19}$ | $\mathbf{63.02}_{\pm 0.30}$ | $78.08_{\pm 0.25}$ | $\mathbf{91.87}_{\pm 0.18}$ | $\mathbf{94.53}_{\pm 0.15}$ | $\mathbf{93.17}_{\pm 0.19}$ |
| IID | SemiFL | $45.86_{\pm 0.12}$ | $62.28_{\pm 0.20}$ | $87.58_{\pm 0.25}$ | $93.01_{\pm 0.15}$ | $94.23_{\pm 0.22}$ | $93.47_{\pm 0.20}$ |
| | FedMatch | $23.05_{\pm 2.40}$ | $31.40_{\pm 2.30}$ | $45.88_{\pm 1.10}$ | $60.05_{\pm 1.10}$ | $71.16_{\pm 2.40}$ | $72.34_{\pm 2.00}$ |
| | FL$^2$ | $33.72_{\pm 1.30}$ | $45.95_{\pm 1.40}$ | $63.15_{\pm 1.10}$ | $78.64_{\pm 1.30}$ | $89.31_{\pm 1.00}$ | $85.36_{\pm 1.10}$ |
| | pFedKnow | $20.25_{\pm 1.60}$ | $33.82_{\pm 2.10}$ | $55.33_{\pm 1.60}$ | $68.06_{\pm 1.20}$ | $62.21_{\pm 2.50}$ | $61.55_{\pm 2.20}$ |
| | **Fed-ARPL** | $\mathbf{53.53}_{\pm 0.20}$ | $\mathbf{63.58}_{\pm 0.10}$ | $\mathbf{88.30}_{\pm 0.18}$ | $\mathbf{93.23}_{\pm 0.17}$ | $\mathbf{95.46}_{\pm 0.16}$ | $\mathbf{93.89}_{\pm 0.18}$ |

freshly computed teacher prototypes $\mathcal{P}_s^t$ with the student consensus to form a pre-momentum global prototype set $\mathcal{P}_{\text{pre}}^{t+1} = \{\mathbf{p}_{c,\text{pre}}^{t+1}\}$:

$$\mathbf{p}_{c,\text{pre}}^{t+1} = \alpha_t \cdot \mathbf{p}_c^s + (1 - \alpha_t) \cdot \bar{\mathbf{p}}_c^t, \tag{7}$$

where the adaptive weight $\alpha_t$ is a decaying function of the round $t$, symbolizing a gradual shift of trust. Crucially, retaining the teacher prototype $\mathbf{p}_c^s$ acts as a stabilizing anchor, preventing the global consensus from drifting excessively due to potentially biased client updates. To ensure temporal stability, a final momentum update is applied:

$$\mathcal{P}_g^{t+1} \leftarrow \beta \cdot \mathcal{P}_g^t + (1 - \beta) \cdot \mathcal{P}_{\text{pre}}^{t+1}, \tag{8}$$

where $\beta$ is the momentum hyperparameter. This reciprocally-enhanced $\mathcal{P}_g^{t+1}$ is then used in the subsequent round for all prototype-based losses, creating a virtuous cycle that enables the framework to converge to a superior global solution. A brief discussion on the communication and privacy implications of prototype exchange is provided in Appendix F.

# 4 EXPERIMENTS

In this section, we conduct comprehensive experiments to validate the effectiveness of our proposed Fed-ARPL framework. We first detail the experimental setup, including the datasets, data partitioning strategies, and baseline methods. We then present the main results, followed by in-depth ablation studies and analyses of our key mechanisms.

## 4.1 EXPERIMENTAL SETUP

**Datasets.** We evaluate our method on three widely-recognized benchmark datasets for semi-supervised learning: CIFAR-10, CIFAR-100 Krizhevsky et al. (2009), and SVHN Netzer et al. (2011). Following the standard "labels-at-server" FSSL protocol Diao et al. (2022); Lee et al. (2024), we simulate scenarios with varying numbers of labeled samples ($N_s$) at the server to assess performance under different degrees of supervision scarcity. Specifically, for CIFAR-10, we test with $N_s \in \{250, 4000\}$; for CIFAR-100, $N_s \in \{2500, 10000\}$; and for SVHN, $N_s \in \{250, 1000\}$. The remaining training data from each dataset is distributed among the clients as unlabeled data.

**Data Partitioning.** To simulate diverse real-world federated environments, we evaluate all methods under both IID and Non-IID data partitioning settings. For the IID setting, the unlabeled data is shuffled and distributed evenly among all clients. For the Non-IID setting, which models statistical heterogeneity, we use a Dirichlet distribution ($\text{Dir}(\alpha)$) to partition the unlabeled data among clients based on their class labels. A smaller $\alpha$ indicates a higher degree of data heterogeneity. In our experiments, we investigate three levels of heterogeneity by setting $\alpha \in \{0.8, 0.5, 0.2\}$, covering a spectrum from mild to severe label distribution skew.

**Implementation Details.** We implement all experiments in PyTorch on NVIDIA GeForce RTX 3090 GPUs. We simulate a federation of $M = 100$ clients with a $10\%$ participation rate per round. The WideResNet-28x2 Zagoruyko & Komodakis (2016) serves as the backbone model for all methods. The total number of communication rounds is 800 (400 for SVHN), with $E = 5$ local epochs for both clients and the server. We use an SGD optimizer (learning rate=0.03, momentum=0.9, weight decay=$5 \times 10^{-4}$). Key techniques include RandAugment Cubuk et al. (2020) for strong augmentation and static Batch Normalization (sBN) to mitigate feature shifts. Baselines are configured following their original papers. To ensure a fair comparison, all models are trained from scratch without using any pre-trained weights. All results are reported as the mean and standard deviation over 3 random seeds. Detailed hyperparameters for all methods are provided in the Appendix.

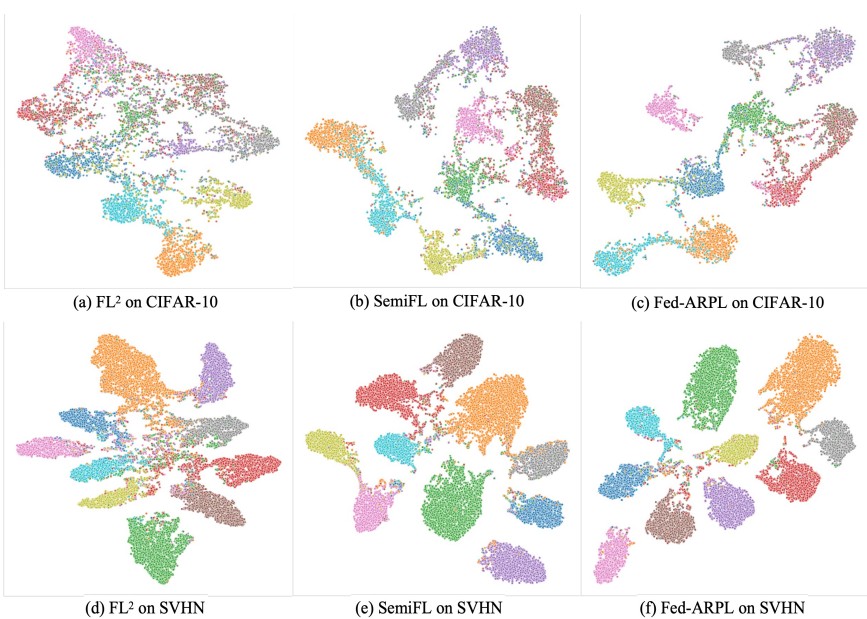

| (a) FL² on CIFAR-10 | (b) SemiFL on CIFAR-10 | (c) Fed-ARPL on CIFAR-10 |
| (d) FL² on SVHN | (e) SemiFL on SVHN | (f) Fed-ARPL on SVHN |

Figure 2: t-SNE visualization of the feature representations learned by different methods on the test sets of (a-c) CIFAR-10 ($N_s = 250$, Dir $\alpha = 0.5$) and (d-f) SVHN ($N_s = 250$, Dir $\alpha = 0.2$). Each point represents a sample, colored by its true class. Fed-ARPL learns significantly more separable and compact class clusters compared to the baselines.

## 4.2 MAIN RESULTS AND ANALYSIS

We evaluate Fed-ARPL against state-of-the-art FSSL methods: FedMatch Jeong et al. (2020), SemiFL Diao et al. (2022), pFedKnow Wang et al. (2023), and (FL)² Lee et al. (2024). Note that supervised prototype methods like FedProto Tan et al. (2022) are excluded as they lack mechanisms for unlabeled data; however, their core principle is implicitly evaluated in our Phase 2 ablation. Table 1 presents the comparison under four distributions and two scarcity levels, demonstrating the consistent superiority of our framework.

**Analysis of Results.** The results in Table 1 comprehensively demonstrate the superiority of Fed-ARPL, which achieves state-of-the-art performance across the vast majority of settings, often by a significant margin. This is particularly evident in challenging scenarios. For instance, on the highly

heterogeneous CIFAR-100 ($N_s = 2500, \text{Dir } \alpha = 0.2$), Fed-ARPL achieves $47.43\%$, surpassing the strong SemiFL baseline by a substantial $4.11\%$. Furthermore, in the extreme label-scarce setting of CIFAR-10 ($N_s = 250, \text{Dir } \alpha = 0.2$), our method attains $69.11\%$ accuracy, outperforming all competitors. This consistent superiority stems from our three-phase strategy: the initial adaptive thresholding and teacher-guided prototyping effectively mitigate the cold-start problem, while the final reciprocal learning phase is crucial for breaking the knowledge bottleneck. We will further dissect the contributions of each phase in subsequent sections.

### 4.3 ABLATION AND ANALYSIS

**Component Contributions.** We first conduct an ablation study to validate the contribution of each component in Fed-ARPL. As shown in Table 2, starting from a SemiFL baseline ($a$), each incrementally added mechanism yields a significant performance gain. Introducing Adaptive Thresholding ($b$) improves the accuracy by 2.13%, confirming its effectiveness against the cold-start

Table 2: Ablation study on the effectiveness of each phase of Fed-ARPL on CIFAR-100 ($N_s = 2500$, Non-IID $\alpha = 0.5$). We report the final average test accuracy (%).

| Framework Configuration | Accuracy (%) |
|---|---|
| ($a$) Baseline (SemiFL w/ fixed $\tau = 0.95$) | 43.53 |
| ($b$) + Adaptive Thresholding (Phase 1) | 45.66 |
| ($c$) + Teacher-Guided Proto. (Phase 2) | 48.78 |
| ($d$) **+ Reciprocal Learning (Fed-ARPL)** | **50.97** |

problem. Subsequently incorporating Teacher-Guided Prototyping ($c$) provides a further 3.12% boost, highlighting the importance of server-guided regularization. Finally, the full Fed-ARPL framework with Reciprocal Learning ($d$) achieves the best performance, affirming that the synergistic combination of all three phases is essential for the framework's superiority.

**Analysis of Learned Feature Representations.** The t-SNE Maaten & Hinton (2008) visualizations in Figure 2 reveal the structural superiority of the feature space learned by Fed-ARPL. Compared to the chaotic and overlapping class distributions of baselines like SemiFL and FL$^2$, Fed-ARPL (c, f) learns remarkably more compact and well-separated class clusters. This robust and discriminative representation is a direct outcome of our guided and reciprocal learning strategy, providing strong visual evidence for the framework's effectiveness.

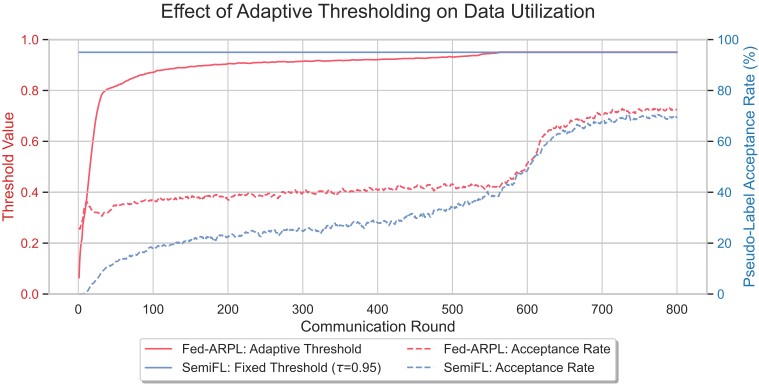

Figure 3: Dynamics of the threshold value (left y-axis, red) and pseudo-label acceptance rate (right y-axis, blue) on CIFAR-100 ($N_s = 2500$, Dir $\alpha = 0.5$). Fed-ARPL's adaptive threshold enables immediate and substantial unlabeled data utilization, while the fixed threshold of the baseline leads to initial data dormancy.

**Mitigating the Cold-Start Problem.** Fed-ARPL's ability to overcome initial learning inertia is mechanistically validated in Figure 3. The visualization shows that our adaptive threshold (red solid line) starts low and rises with the model's proficiency. This directly enables a high pseudo-label acceptance rate of around 40% from the very first rounds (red dashed line). In stark contrast, the baseline's static, high threshold keeps its acceptance rate near zero for the initial 200 rounds. This

immediate and substantial utilization of unlabeled client data provides a significant head start, which, as evidenced by our main results, translates into faster convergence and superior final performance.

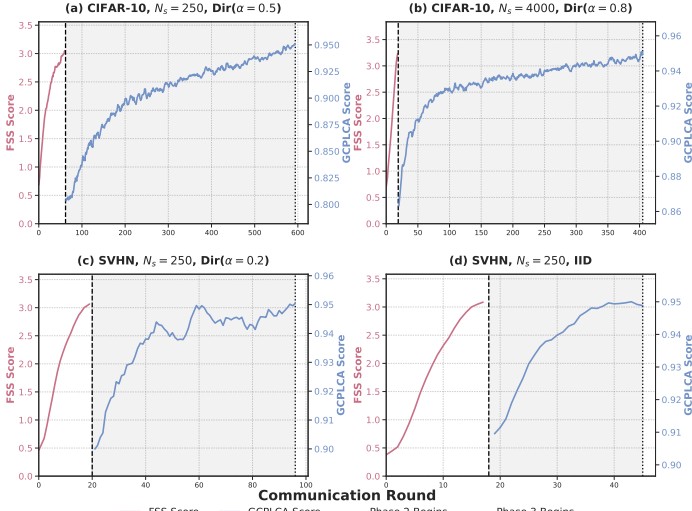

Figure 4: Dynamics of the automated phase transition mechanism. Each subplot shows the FSS score (left y-axis, red) and GCPLCA score (right y-axis, blue). The dashed and dotted vertical lines mark the data-driven transitions into Phase 2 and Phase 3, respectively. The shaded backgrounds visually distinguish the three learning phases.

**Framework Adaptivity and Robustness.** A core tenet of Fed-ARPL is its ability to operate autonomously. Figure 4 provides a transparent view of this system, showing that the transitions between phases are not pre-programmed heuristics but are direct, logical consequences of the system achieving key learning milestones, triggered when the FSS and GCPLCA scores reach stable plateaus. Furthermore, we analyze the sensitivity of our key hyperparameters in Figure 5. The results demonstrate that Fed-ARPL is robust across a reasonable range of values for its core parameters. A detailed breakdown and discussion for each hyperparameter is provided in Appendix.

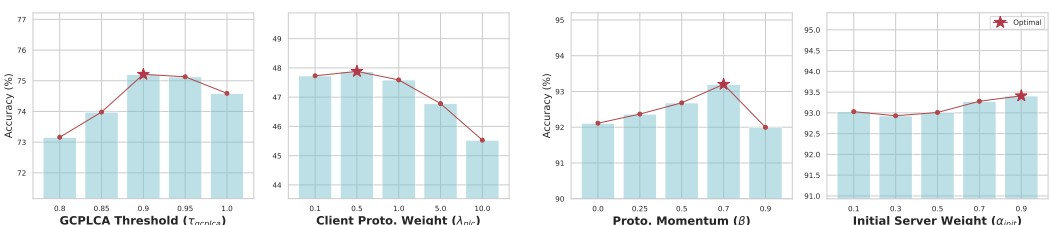

Figure 5: Analysis of key hyperparameters in Fed-ARPL. We vary one hyperparameter at a time while keeping others at their optimal values. The star ($\star$) indicates the selected setting for each parameter. Each subplot corresponds to a specific experimental setting detailed in the text.

## 5 CONCLUSION

In this paper, we introduced Fed-ARPL, a novel framework that addresses the fundamental challenges of cold-start inertia and the knowledge bottleneck in Federated Semi-Supervised Learning. Fed-ARPL reconceptualizes the learning process as a dynamic, three-phase pedagogical journey. Our framework initially dismantles the learning impasse through a synergistic application of adaptive thresholding and one-way prototypical guidance. It then culminates in a reciprocal learning phase, where student-feedback via high-quality client prototypes shatters the server's performance ceiling. Comprehensive experiments validate that Fed-ARPL significantly outperforms state-of-the-art baselines, establishing a new benchmark for efficient and effective FSSL.

## ETHICS STATEMENT

Our work investigates Federated Semi-Supervised Learning (FSSL), a paradigm designed to enable collaborative machine learning while preserving data privacy. We believe this research can contribute positively to the responsible development of AI by allowing powerful models to be trained on decentralized data without requiring clients to share their sensitive, raw information. This can unlock beneficial applications in domains where data is private and labeled examples are scarce, such as in IoT, healthcare and finance, thus promoting a more accessible and data-efficient approach to AI.

Nevertheless, we acknowledge potential negative impacts. A model trained via FSSL, though privacy-preserving in its process, could still be deployed for malicious purposes. More specific to our setting, biases present in the central server's small guiding dataset or within the distributed, unlabeled client data could be propagated and potentially amplified across the entire federation. Our experiments are conducted exclusively on publicly available benchmark datasets (CIFAR-10, CIFAR-100, and SVHN), which contain no personally identifiable information. However, we recognize that biases inherent in these datasets could still influence model behavior. We encourage further research on fairness, bias detection, and mitigation techniques specifically designed for the unique challenges of the FSSL setting.

Overall, we emphasize that our framework is intended for beneficial applications where data privacy is paramount. We explicitly discourage its use in ways that could lead to unfair or biased automated decision-making, compromise user privacy despite the federated setup, or be used for surveillance or other malicious activities that harm individuals or society.

## REPRODUCIBILITY STATEMENT

We are committed to ensuring the reproducibility of our work. All datasets used in our experiments are publicly available: CIFAR-10 and CIFAR-100 (Krizhevsky et al., 2009), and SVHN (Netzer et al., 2011). We will release our full source code, trained model checkpoints, and configuration files upon acceptance of the manuscript. The code will include scripts for data partitioning, training, and evaluation to facilitate the verification of our results. Our source code will be submitted along with the supplementary materials.

We describe all necessary implementation details in Section 4 of the main paper. This includes the model architecture (WideResNet-28x2 (Zagoruyko & Komodakis, 2016)), optimization settings (SGD optimizer with learning rate, momentum, and weight decay), data partitioning strategies (IID and Non-IID Dirichlet distribution), and data preprocessing pipelines (RandAugment for strong augmentation and static Batch Normalization). All experiments were repeated using 3 different random seeds to ensure consistent and statistically valid results across runs.

Our experiments were conducted using PyTorch on NVIDIA GeForce RTX 3090 GPUs. All reported results in tables and figures can be fully reproduced using our released code and configuration files. We will also provide scripts and notebooks to regenerate the main figures and tables directly from the trained checkpoints to facilitate verification and reuse by the community.

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

## A LLM USAGE

In the preparation of this manuscript, we employed Large Language Models (LLMs), including ChatGPT 4o, as sophisticated auxiliary tools under our direct supervision. Our use of these models was confined to specific aspects of manuscript improvement to enhance its quality and readability. First, we utilized LLMs for language polishing and grammar correction. All suggested modifications were carefully reviewed and implemented by the authors to ensure that the original scientific meaning, core arguments, and contributions remained unchanged. Second, LLMs provided significant assistance in optimizing the paper's typesetting, particularly in refining complex LaTeX code for tables with varied formatting and figures requiring specific layouts. The final implementation of all formatting and layout decisions was manually verified and adjusted by the authors to ensure adherence to the conference style and to improve the overall presentation. In all instances, these LLMs served as productivity aids, while the conceptualization, scientific methodology, experimental results, and final conclusions of this work are entirely our own.

## B DETAILED PRELIMINARIES OF FSSL AND PROTOTYPICAL LEARNING

This section provides a detailed formulation of the Federated Semi-Supervised Learning (FSSL) and Prototypical Learning concepts that form the foundation of our work. These formulations are referenced in the algorithm descriptions.

**FSSL Baseline Workflow.** A typical FSSL communication round $t$, inspired by the alternating training procedure in SemiFL Diao et al. (2022), unfolds as follows:

SERVER TRAINING AND DISTRIBUTION. The server first updates its model parameters from $\theta_s^t$ using its labeled data $\mathcal{D}_s$. Specifically, it performs several local epochs to minimize the supervised cross-entropy loss:

$$\mathcal{L}_s = -\frac{1}{|\mathcal{B}_s|} \sum_{(x,y) \in \mathcal{B}_s} \log p_{\theta_s^t}(y|x), \tag{9}$$

where $\mathcal{B}_s$ is a mini-batch sampled from $\mathcal{D}_s$. The resulting model parameters are then designated as the global model for the current round, $\theta_g^t \leftarrow \theta_s^t$.

CLIENT PSEUDO-LABELING AND TRAINING. Upon receiving the global model $\theta_g^t$, each client $C_k$ leverages it to generate pseudo-labels for its local unlabeled data. The client then updates its local model by minimizing a consistency regularization loss, $\mathcal{L}_k^u$, which is formally defined in the main text in Eq. equation 1.

SERVER AGGREGATION. After local training, clients upload their updated models $\{\theta_k^t\}$. The server then aggregates these models using Federated Averaging McMahan et al. (2017):

$$\theta_s^{t+1} \leftarrow \sum_{k \in \mathcal{C}_{sel}^t} \frac{|\mathcal{D}_k^u|}{|\mathcal{D}_{sel}^u|} \theta_k^t, \tag{10}$$

where $|\mathcal{D}_{sel}^u| = \sum_{j \in \mathcal{C}_{sel}^t} |\mathcal{D}_j^u|$.

**Prototypical Learning.** A prototype $\mathbf{p}_c$ for class $c$ is the mean feature embedding of samples belonging to that class. The general form of the prototype consistency loss, which regularizes the feature extractor, is:

$$\mathcal{L}_{proto} = \|h_\phi(x) - \mathbf{p}_y\|_2^2. \tag{11}$$

## C EXPERIMENT DETAILS

### C.1 DATASET DETAILS

We validate the performance of our Fed-ARPL framework on three standard benchmark datasets widely used in semi-supervised and federated learning literature. For all datasets, we adhere to

the "labels-at-server" FSSL setting. A small, class-balanced subset of the original training set is designated as the server's labeled data $\mathcal{D}_s$, while the entire remaining portion of the training set is partitioned among the 100 clients as unlabeled data $\mathcal{D}_k^u$. All images are resized or cropped to a standard resolution of $32 \times 32$ pixels.

**CIFAR-10** The CIFAR-10 dataset Krizhevsky et al. (2009) consists of 60,000 $32 \times 32$ color images in 10 distinct object classes (e.g., 'airplane', 'automobile', 'bird'). The official split provides 50,000 training images and 10,000 test images. In our experiments, we create the server's labeled set $\mathcal{D}_s$ by randomly sampling either 25 or 400 images per class from the training set, resulting in a total of $N_s = 250$ or $N_s = 4000$ labeled samples, respectively.

**CIFAR-100** The CIFAR-100 dataset Krizhevsky et al. (2009) is a more challenging benchmark, containing the same number of images as CIFAR-10 but distributed across 100 fine-grained classes. These 100 classes are further grouped into 20 superclasses. For our experiments, which focus on the 100-class classification task, we construct the server's labeled set $\mathcal{D}_s$ by sampling either 25 or 100 images per class from the training set, leading to a total of $N_s = 2500$ or $N_s = 10000$ labeled samples.

**SVHN** The Street View House Numbers (SVHN) dataset Netzer et al. (2011) is a real-world image dataset obtained from house numbers in Google Street View images. It contains over 600,000 digit images, with 73,257 digits for training and 26,032 for testing. The task is to classify the central digit in each $32 \times 32$ color image into one of 10 classes (digits 0-9). For our FSSL setting, we form the server's labeled set $\mathcal{D}_s$ by sampling either 25 or 100 images per class from the training set, resulting in $N_s = 250$ or $N_s = 1000$ labeled samples. The slightly different number of total samples for SVHN compared to CIFAR-10 (i.e., $N_s = 250$ instead of 1000) is chosen to align with established benchmarks in the FSSL literature Diao et al. (2022); Lee et al. (2024).

## C.2 DETAILS OF LEARNING SETUP

**Implementation Details.** All experiments are conducted using PyTorch on NVIDIA RTX 3090 and A10 GPUs. For a fair and controlled comparison, all baseline methods were re-implemented within our unified FSSL framework, and we adhered to the hyperparameter settings specified in their original papers where applicable. Table 3 provides a comprehensive summary of the key hyperparameters used throughout our experiments.

For our proposed Fed-ARPL, several key hyperparameters govern its phased learning dynamics. The client-side prototype loss weight, $\lambda_{plc}$, is set to 1.0 for CIFAR-10/SVHN and increased to 10.0 for the more complex CIFAR-100 to provide stronger regularization. The server-side weight, $\lambda_{pls}$, is kept at 1.0. The temporal stability of global prototypes is maintained by a momentum term, $\beta = 0.7$.

In the reciprocal learning phase, the initial weight for the server's own prototypes, $\alpha_0$, is set to 0.7, which then decays to gradually shift trust to the client consensus. The automated phase transitions are controlled by stability criteria over a moving window of $W_{fss} = W_{gcplca} = 5$ rounds. The transition to Phase 2 requires the FSS score to be consistently high and stable (mean $\tau_{fss} = 3.0$, std $\sigma_{fss} = 0.3$), while the transition to Phase 3 requires the GCPLCA score to meet similar criteria (mean $\tau_{gcplca} = 0.95$, std $\sigma_{gcplca} = 0.05$).

Finally, for client-side feedback generation in Phase 3, a stricter threshold of $\tau' = \tau^t + 0.05$ is used to ensure the quality of local prototypes.

## C.3 ANALYSIS OF KEY HYPERPARAMETERS

**Robustness and Unified Configuration.** While Fed-ARPL introduces a set of hyperparameters to govern its phase transitions and regularization strengths, our extensive empirical evaluation demonstrates that the framework is remarkably robust to these choices. Notably, we employed a **unified configuration** for the majority of these parameters (e.g., transition windows $W = 5$, stability thresholds $\sigma_{fss} = 0.3$, $\sigma_{gcplca} = 0.05$) across all disparate datasets (CIFAR-10, CIFAR-100, SVHN) and heterogeneity settings (from IID to Dir $\alpha = 0.2$). The fact that Fed-ARPL consistently achieves SOTA performance without dataset-specific fine-tuning of these thresholds suggests that they are not brittle. Although manual calibration is not strictly required for strong performance, we acknowledge

| Scope | Hyperparameter | Value |
|---|---|---|
| Global | Communication Rounds | 800 |
| | Client Participation Rate ($C$) | 0.1 |
| | Optimizer | SGD |
| | FedAvg Momentum | 0.5 |
| Server & Client | Local Epochs ($E$) | 5 |
| | Learning Rate ($\eta$) | 0.03 |
| | Weight Decay | $5 \times 10^{-4}$ |
| | Momentum | 0.9 |
| Client-Side SSL | Unlabeled Batch Size ($B_u$) | 32 (for FL$^2$), 10 (for others) |
| | Strong Augmentation | RandAugment ($N = 2, M = 10$) |
| **Fed-ARPL Specific** | Proto. Loss Weight (Client, $\lambda_{plc}$) | 1.0 |
| | Proto. Loss Weight (Server, $\lambda_{pls}$) | 1.0 |
| | Proto. Momentum ($\beta$) | 0.7 |
| | Reciprocal Update Weight (Initial, $\alpha_0$) | 0.7 |
| | Phase 1 Trans. Window ($W_{fss}$) | 5 |
| | Phase 1 Trans. Mean Threshold ($\tau_{fss}$) | 3.0 |
| | Phase 1 Trans. Std. Threshold ($\sigma_{fss}$) | 0.3 |
| | Phase 2 Trans. Window ($W_{gcplca}$) | 5 |
| | Phase 2 Trans. Mean Threshold ($\tau_{gcplca}$) | 0.95 |
| | Phase 2 Trans. Std. Threshold ($\sigma_{gcplca}$) | 0.05 |

Table 3: Hyperparameters used in our experiments.

that exploring self-adaptive mechanisms—such as dynamically inferring transition points based on the rate of change of metric values—represents a promising avenue for future work to further enhance the framework's "plug-and-play" nature.

Following this general robustness, we conduct a specific sensitivity analysis on four key hyperparameters to illustrate their impact dynamics. As shown in Figure 5, our framework maintains high performance within a reasonable range of values.

**GCPLCA Threshold** ($\tau_{gcplca}$). This parameter controls the transition to Phase 3. The optimal performance on CIFAR-10 is achieved at $\tau_{gcplca} = 0.9$. Lower values risk incorporating noisy prototypes prematurely, while higher values conservatively delay the beneficial student-feedback mechanism.

**Client Prototype Weight** ($\lambda_{plc}$). This weight balances local learning with global guidance on the challenging CIFAR-100 dataset. A value of $0.5$ is optimal, as smaller values offer insufficient regularization against heterogeneity, while larger values stifle local learning.

**Prototype Momentum** ($\beta$). This parameter stabilizes prototype evolution on the rapidly converging SVHN dataset. The optimum at $\beta = 0.7$ effectively smooths update volatility without hindering the model's responsiveness to new knowledge.

**Initial Server Weight** ($\alpha_{init}$). This parameter sets the initial trust balance in Phase 3 on CIFAR-10 (IID). The optimum at $\alpha_{init} = 0.9$ suggests that a strong initial anchor to the teacher's reliable knowledge is beneficial for a stable transition into the reciprocal phase, before the weight decays to favor client insights.

### C.3.1 ANALYSIS OF THE ADAPTIVE PHASE TRANSITION MECHANISM

To provide a transparent view into this "auto-pilot" system, Figure 4 visualizes the evolution of our two key transition metrics—the Feature-Space Separation (FSS) score and the Global Client Pseudo-Label Confidence Aggregation (GCPLCA) score—across four diverse experimental settings.

The figure provides a clear, mechanistic proof of our framework's adaptive nature. In each subplot, the learning process begins in Phase 1 (white background), where the FSS score (red curve) rapidly increases as the server's feature extractor learns a more structured representation. The transition to Phase 2 (light gray background) is triggered precisely when the FSS score begins to plateau, signifying that the "teacher" is ready to provide stable guidance. This causal link is evident across all scenarios, from the fast transition in the high-data SVHN IID setting (d) to the more measured progression in the data-scarce, heterogeneous CIFAR-10 setting (a).

Upon entering Phase 2, the GCPLCA score (blue curve) begins its ascent, indicating that clients, under the teacher's prototypical guidance, are becoming progressively more confident in their pseudo-labels. The final transition to Phase 3 (light blue background, not reached in all plots) occurs only after the GCPLCA score reaches a sustained high level of stability. This visualization unequivocally demonstrates that the phase transitions in Fed-ARPL are not pre-programmed heuristics but are direct, logical consequences of the system achieving key learning milestones, ensuring that each mechanism is deployed precisely when it is most beneficial.

---

**Algorithm 1** The Fed-ARPL Framework

---

**Input:** Labeled server data $\mathcal{D}_s$; Unlabeled client data $\{\mathcal{D}_k^u\}_{k=1}^M$; Number of communication rounds $T$; Number of local epochs $E$; Client participation rate $C$.
**Parameter:** Local learning rate $\eta$; Prototype loss weight $\lambda_{plc}$; Stricter threshold $\tau'$; Adaptive weight $\alpha_t$; Prototype momentum $\beta$.
**Initialize:** Server model $\theta_s^0$; Global prototypes $\mathcal{P}_g^0 = \emptyset$; System phase $\texttt{Phase} \leftarrow 1$; Histories $\mathbf{H} = \{\mathcal{H}_{fss}, \mathcal{H}_{gcplca}\} \leftarrow \emptyset$.

1: **System executes:**
2: **for** each communication round $t = 1, 2, \ldots, T$ **do**
3: $\quad \theta_g^t \leftarrow \texttt{ServerModelUpdate}(\mathcal{D}_s, \theta_s^t, \mathcal{P}_g^t, \texttt{Phase})$
4: $\quad$ Select a random client subset $\mathcal{C}_{sel}^t \subset \mathcal{C}$
5: $\quad$ **for** each client $C_k \in \mathcal{C}_{sel}^t$ **in parallel do**
6: $\quad\quad \{\theta_k^t, \mathcal{P}_k^t, CPLCA_k\} \leftarrow \texttt{ClientUpdate}(\mathcal{D}_k^u, \theta_g^t, \mathcal{P}_g^t, \texttt{Phase})$
7: $\quad$ **end for**
8: $\quad$ Let $\mathbf{P}^t = \{\mathcal{P}_k^t\}_{k \in \mathcal{C}_{sel}^t}$, $\mathbf{C}^t = \{CPLCA_k\}_{k \in \mathcal{C}_{sel}^t}$
9: $\quad$ Aggregate client models: $\theta_s^{t+1} \leftarrow \sum_{k \in \mathcal{C}_{sel}^t} \frac{|\mathcal{D}_k^u|}{|\mathcal{D}_{sel}^u|} \theta_k^t$
10: $\quad$ Update protos & phase: $\mathcal{P}_g^{t+1}, \texttt{Phase} \leftarrow \texttt{ServerProtoUpdate}(\mathbf{P}^t, \mathbf{C}^t, \theta_g^t, \mathcal{P}_g^t, \texttt{Phase}, \mathbf{H})$
11: **end for**

---

# D ALGORITHM

*For brevity in the algorithmic presentation, we denote the collection of client prototypes as $\mathbf{P}^t$, the collection of client confidence scores as $\mathbf{C}^t$, and the set of phase-transition histories as $\mathbf{H}$.*

---

**Algorithm 2** Server-Side Model Update

---

1: **function** $\texttt{ServerModelUpdate}(\mathcal{D}_s, \theta_s, \mathcal{P}_g, \texttt{Phase})$
2: **for** each local epoch $e = 1, \ldots, E$ **do**
3: $\quad$ **for** batch $(x, y) \in \mathcal{D}_s$ **do**
4: $\quad\quad$ Calculate supervised loss $\mathcal{L}_s$ via Eq. equation 9
5: $\quad\quad$ **if** $\texttt{Phase} = 3$ **then**
6: $\quad\quad\quad$ Calculate prototype loss $\mathcal{L}_{proto}$ for the batch
7: $\quad\quad\quad$ $\mathcal{L}_s \leftarrow \mathcal{L}_s + \lambda_{pls} \cdot \mathcal{L}_{proto}$
8: $\quad\quad$ **end if**
9: $\quad\quad$ $\theta_s \leftarrow \theta_s - \eta \nabla_{\theta_s} \mathcal{L}_s$
10: $\quad$ **end for**
11: **end for**
12: **return** $\theta_s$

---

---

**Algorithm 3** Server-Side Proto & Phase Update

---

1: **function** ServerProtoUpdate($\mathbf{P}, \mathbf{C}, \theta_g, \mathcal{P}_g,$
                                 Phase, $\mathbf{H}$)
2:   Let $\mathcal{H}_{fss}, \mathcal{H}_{gcplca} \leftarrow \mathbf{H}$
3:   Compute server prototypes $\mathcal{P}_s$ from $\mathcal{D}_s$ using $h_{\phi_g}$
4:   **if** Phase $= 1$ **then**
5:     Calculate FSS and update $\mathcal{H}_{fss}$
6:     **if** Transition criterion met (Sec 4.1) **then**
7:       Phase $\leftarrow 2$
8:     **end if**
9:   **else if** Phase $= 2$ **then**
10:    Calculate GCPLCA $= \text{mean}(\mathbf{C})$ and update $\mathcal{H}_{gcplca}$
11:     **if** Transition criterion met (Sec 4.2) **then**
12:       Phase $\leftarrow 3$
13:     **end if**
14:   **end if**
15:   **if** Phase $< 3$ **then**
16:     $\mathcal{P}_{pre} \leftarrow \mathcal{P}_s$
17:   **else**
18:     Aggregate client prototypes $\bar{\mathcal{P}} \leftarrow \text{mean}(\mathbf{P})$
19:     Form enriched prototypes $\mathcal{P}_{pre}$ via Eq. equation 7
20:   **end if**
21:   Update global prototypes $\mathcal{P}_g^{new}$ via Eq. equation 8
22:   **return** $\mathcal{P}_g^{new}$, Phase

---

**Overall Procedure.** Algorithm 1 outlines the main, high-level procedure of the Fed-ARPL framework. The system operates over a series of communication rounds $T$. At the beginning of each round $t$, the server first performs its local training and updates its model to produce the global model $\theta_g^t$ for distribution (Line 3), with the detailed procedure described in Algorithm 2. After broadcasting $\theta_g^t$ to a selected subset of clients $\mathcal{C}_{sel}^t$, each client executes its local update in parallel (Line 6). This core client-side operation, which adapts based on the current system phase, is encapsulated in Algorithm 4. Upon receiving the updated models $\{\theta_k^t\}$ and other feedback from clients, the server first aggregates the model parameters via Federated Averaging (Line 8). The round concludes with a crucial step where the server updates the global prototypes and determines if a phase transition is warranted (Line 9). This final, adaptive step is detailed in Algorithm 3.

**Server Model Training.** Algorithm 2 details the server's local model training process. The primary objective is to leverage the small, clean labeled dataset $\mathcal{D}_s$ to refine the model parameters. The function takes the current server model $\theta_s$, the global prototypes $\mathcal{P}_g$, and the current phase as input. For a set number of local epochs $E$, the server minimizes the standard supervised cross-entropy loss as defined in Eq. equation 9. A key aspect of our reciprocal learning paradigm is introduced in Phase 3 (Line 5): the server's training objective is augmented with a prototype consistency loss, compelling its own feature extractor to align with the enriched global prototypes. This ensures that the "teacher" also learns from the "student feedback" embedded in $\mathcal{P}_g$, thus closing the reciprocal knowledge loop. The function returns the updated model parameters, which are then used as the global model for the current round.

**Server-Side Adaptation.** Algorithm 3 encapsulates the core intelligence and adaptability of the Fed-ARPL framework. This function is responsible for both managing the evolution of the global prototypes and governing the transition between learning phases. It begins by computing the server's own prototypes $\mathcal{P}_s$ from its labeled data (Line 2). Subsequently, it performs phase-specific checks: in Phase 1, it evaluates the FSS score to test for feature space maturity (Lines 4-8), while in Phase 2, it evaluates the aggregated client confidence (GCPLCA) to determine client proficiency (Lines 9-13). The global prototype update logic is also phase-dependent. Before Phase 3, the global consensus is guided solely by the server's knowledge (Line 15). Once in Phase 3, the server integrates the aggregated client prototypes ($\bar{\mathcal{P}}$) with its own, forming an enriched consensus via Eq. equation 7

---

**Algorithm 4** Client-Side Local Update

---

1: **function** ClientUpdate($\mathcal{D}_k^u, \theta_g, \mathcal{P}_g,$ Phase)
2: $\theta_k \leftarrow \theta_g$
3: Calculate adaptive threshold $\tau$ via Eq. equation 2
4: Calculate $CPLCA_k = \mathbb{E}_{x \in \mathcal{D}_k^u}[\max(p_{\theta_g}(y|x))]$
5: **for** each local epoch $e = 1, \ldots, E$ **do**
6:    **for** batch $x_j \in \mathcal{D}_k^u$ **do**
7:       Calculate consistency loss $\mathcal{L}_k^u$ via Eq. equation 1 using $\tau^t$
8:       **if** Phase $\geq 2$ **then**
9:          Calculate prototype loss $\mathcal{L}_{proto}$ via Eq. equation 5
10:          $\mathcal{L}_k^u \leftarrow \mathcal{L}_k^u + \lambda_{plc} \cdot \mathcal{L}_{proto}$
11:       **end if**
12:       $\theta_k \leftarrow \theta_k - \eta \nabla_{\theta_k} \mathcal{L}_k^u$
13:    **end for**
14: **end for**
15: $\mathcal{P}_k \leftarrow \emptyset$
16: **if** Phase $= 3$ **then**
17:    Compute local prototypes $\mathcal{P}_k$ from high-confidence samples
18: **end if**
19: **return** $\theta_k, \mathcal{P}_k, CPLCA_k$

---

(Lines 18-19). Finally, a momentum update (Eq. equation 8) is applied to ensure temporal stability before the new prototypes are returned (Line 21).

**Client Local Update.** Algorithm 4 describes the complete procedure executed by each client. Upon receiving the global model $\theta_g$, the client first performs two preparatory steps: it computes the server-dictated adaptive threshold $\tau$ for the current round (Line 3) and calculates its own confidence score $CPLCA_k$ for reporting back to the server (Line 4). The core of the local training (Lines 5-13) involves minimizing a loss function that adapts with the system phase. In all phases, it uses the adaptive consistency loss from Eq. equation 1. From Phase 2 onwards, this loss is augmented with the prototype consistency loss (Eq. equation 5) to align with the global prototypes (Lines 9-11). The client's role as a knowledge contributor is activated only in Phase 3 (Lines 15-17), where it computes and prepares its local prototypes $\mathcal{P}_k$ from a high-confidence subset of its data. Finally, the client returns its updated model, its computed local prototypes (which are empty if not in Phase 3), and its confidence score to the server.

# E  ON THE DESIGN OF PHASE TRANSITION METRICS

In this section, we provide a detailed justification for our design choices regarding the FSS and GC-PLCA metrics, addressing their theoretical grounding, comparison to alternatives, and the rationale behind their hyperparameterization.

**Theoretical Grounding and Design of FSS.** The Feature-Space Separation (FSS) score is designed to be a direct and computationally efficient measure of the geometric quality of the learned representations. While standard clustering validity indices such as the Silhouette score Rousseeuw (1987) or the soft davies-bouldin separation measure Vergani & Binaghi (2018) exist, we specifically designed FSS to be more aligned with our framework's internal logic and the constraints of the federated setting. Our choice is motivated by two key factors. First, FSS is prototype-centric; as our entire framework revolves around prototypes, FSS directly evaluates the quality of the feature space with respect to the prototypes themselves, which is more coherent with our method's architecture than general-purpose metrics that require pairwise distance calculations across all samples. Second, FSS is highly efficient; its complexity scales with the number of classes ($O(K^2)$) rather than the number of samples, making it scalable for large datasets. This is in stark contrast to metrics like the Silhouette score, whose reliance on all-pairs sample distances can be computationally prohibitive in a federated context.

The design of FSS is also deeply rooted in the principles of deep metric learning Kaya & Bilge (2019). A primary goal of training a deep classifier is to learn an embedding space where data is semantically organized. Seminal works like Prototypical Networks Snell et al. (2017) and those employing Triplet Loss Schroff et al. (2015) formalize this objective as simultaneously minimizing intra-class variance and maximizing inter-class separation. Our FSS score (Eq. 3) is a direct, closed-form quantification of this dual objective. Its numerator, which measures the distance to the nearest neighboring class prototype, quantifies inter-class separation, while its denominator, which measures the average sample-to-prototype distance, quantifies intra-class compactness.

**Theoretical Grounding of GCPLCA.** The GCPLCA score is designed to be a lightweight yet effective proxy for the collective maturity and reliability of the client models across the federation.

*Theoretical Motivation:* The use of GCPLCA is grounded in the extensive literature on uncertainty estimation in deep learning. The maximum softmax probability (MSP) of a model's prediction is widely used as a practical, albeit imperfect, proxy for its predictive confidence and, inversely, its epistemic uncertainty, the uncertainty arising from a lack of knowledge Guo et al. (2017). In the context of semi-supervised learning, this principle is the very foundation of confidence-based pseudo-labeling, validated by seminal works like FixMatch Sohn et al. (2020). Our GCPLCA metric elevates this principle from the level of individual samples to the macro-level of the entire federated system. A high and stable GCPLCA indicates that the collective of client models has reached a state of low epistemic uncertainty, suggesting they have converged to a consistent state where their pseudo-label-driven knowledge is reliable enough to be shared.

**On Hyperparameter Tuning and Self-Adaptation.** Our current framework uses fixed, empirically chosen thresholds for the stability criteria (e.g., $\tau_{fss}, \sigma_{fss}$). We acknowledge that this is a limitation and that these values may require some tuning for vastly different datasets or model architectures. However, our sensitivity analyses in the main paper (if you have them, refer to them here) show that the framework is robust to moderate variations in these parameters.

The suggestion to explore adaptive or self-tuning mechanisms is a highly valuable direction for future work. For instance, the FSS/GCPLCA thresholds could potentially be made dynamic. One could imagine a meta-learning approach or a simple heuristic where the transition is triggered not by a fixed value, but when the *rate of change* (i.e., the first derivative) of the FSS or GCPLCA score falls below a certain threshold for a sustained period. This would signify that the metric has entered a plateau, indicating a natural point for a phase transition. We have added this promising direction to our future work section in the main paper.

# F System Overhead and Privacy Analysis

In Phase 3 of Fed-ARPL, clients upload both model parameters and local prototypes. We provide a comprehensive analysis of the associated communication, computation, and privacy considerations, directly addressing the trade-offs of this design choice.

**Communication Overhead Analysis.** Our framework introduces prototype exchange in Phases 2 and 3. We provide a detailed, phase-by-phase analysis of the round-trip communication overhead. The total cost per round for a selected client is the sum of its downlink (server-to-client) and uplink (client-to-server) payloads. The two main components are the model parameters ($S_{model}$) and the prototypes ($S_{proto}$).

For our experimental setup using the WideResNet-28x2 backbone ($D_{feat} = 128$) and the CIFAR-100 dataset ($K = 100$), the component sizes are:

- **Model Size ($S_{model}$):** $\approx 1.47$ million parameters $\times$ 4 bytes/param $\approx$ **5.88 MB**.

- **Prototype Size ($S_{proto}$):** 100 classes $\times 128$ dimensions $\times$ 4 bytes/dim $\approx$ **51.2 KB**.

We now analyze the communication cost per selected client in each phase:

- **Phase 1 (Baseline):** The server sends the model, and the client sends back the updated model. *Downlink:* $S_{model}$. *Uplink:* $S_{model}$. *Total:* $2 \times S_{model} \approx 11.76$ MB.

- **Phase 2 (Teacher-Guided):** The server sends both the model and the global prototypes. The client sends back only the model.
  *Downlink: $S_{model} + S_{proto}$. Uplink: $S_{model}$. Total: $2 \times S_{model} + S_{proto}$.*
  The additional overhead compared to Phase 1 is $S_{proto}$, which constitutes an increase of only $\frac{S_{proto}}{2 \times S_{model}} = \frac{51.2\,\text{KB}}{11.76\,\text{MB}} \approx \mathbf{0.44\%}$.

- **Phase 3 (Reciprocal):** The server sends the model and global prototypes. The client sends back the model and its local prototypes.
  *Downlink: $S_{model} + S_{proto}$. Uplink: $S_{model} + S_{proto}$. Total: $2 \times S_{model} + 2 \times S_{proto}$.*
  The additional overhead compared to Phase 1 is $2 \times S_{proto}$, resulting in a total increase of $\frac{2 \times S_{proto}}{2 \times S_{model}} = \frac{102.4\,\text{KB}}{11.76\,\text{MB}} \approx \mathbf{0.87\%}$.

This detailed analysis demonstrates that even in the most communication-intensive phase (Phase 3), the total round-trip communication overhead increases by less than 1%. This marginal cost is an insignificant price for the substantial accuracy gains achieved by our adaptive and reciprocal learning mechanisms, confirming a highly favorable utility-cost trade-off.

**Computation and Latency.** The additional computational cost of Fed-ARPL is also minimal and primarily incurred during two steps:

1. **Local Prototype Computation (Client):** This involves a single forward pass over a high-confidence subset of the client's data to extract features, followed by a simple averaging operation. This is computationally inexpensive compared to the multiple epochs of gradient-based training already performed.

2. **Prototype Aggregation (Server):** The server performs a weighted average of the received prototypes. This is a simple, non-iterative arithmetic operation whose latency is negligible compared to the time required for model aggregation and subsequent server-side training.

In summary, the additional computations are non-iterative and significantly less intensive than the core training loop, leading to no discernible increase in overall training time per round compared to baselines like SemiFL, as both run for identical local epochs.

**Privacy Considerations.** The introduction of prototype exchange warrants a careful consideration of privacy implications. We analyze this from two perspectives: the inherent risk and potential mitigations.

*Inherent Risk Analysis:* Uploading prototypes, which are mean feature embeddings, does introduce an information channel that is absent in model-only FL. While prototypes do not leak raw data, they can leak distributional information about a client's local data statistics (e.g., the presence or absence of certain classes, or the feature-space centroid of a class). However, it is crucial to contextualize this risk. In our "labels-at-server" setting, the server already possesses knowledge of the global class set. The primary new information leaked by a prototype $\mathbf{p}_c^k$ is the specific feature distribution of class $c$ on client $k$.

When the server possesses both the local model update ($\Delta\theta_k$) and the local prototypes ($\mathcal{P}_k$), the potential for sophisticated inference attacks theoretically increases. An adversary at the server could potentially use the prototypes as "anchors" to better interpret the gradients in $\Delta\theta_k$, possibly improving the fidelity of gradient-based reconstruction attacks Zhu et al. (2019). However, we argue that this increased risk is marginal in our specific context. The prototypes are computed only on pseudo-labeled data and represent an average over many samples, which inherently provides a level of aggregation-based privacy.

*Potential Mitigations:* Addressing this potential risk is an active and important research area. Our framework is compatible with existing privacy-enhancing technologies (PETs). Potential countermeasures, which are orthogonal to our core contribution, include:

- Applying Differential Privacy (DP) Abadi et al. (2016) to the local prototypes before uploading. This involves adding calibrated noise to the prototype vectors to provide formal privacy guarantees.

- Employing Secure Aggregation Bonawitz et al. (2017) protocols. In this setup, the server would only receive the aggregated "student consensus" prototype ($\bar{\mathcal{P}}^t$) and would be unable to access any individual client's prototype ($\mathcal{P}_k^t$), completely mitigating any server-side inference risk based on individual prototypes.

A formal investigation of these privacy-preserving extensions and their impact on the utility of our reciprocal learning mechanism is a promising direction for future work.

## G  MORE EXPERIMENTAL RESULTS

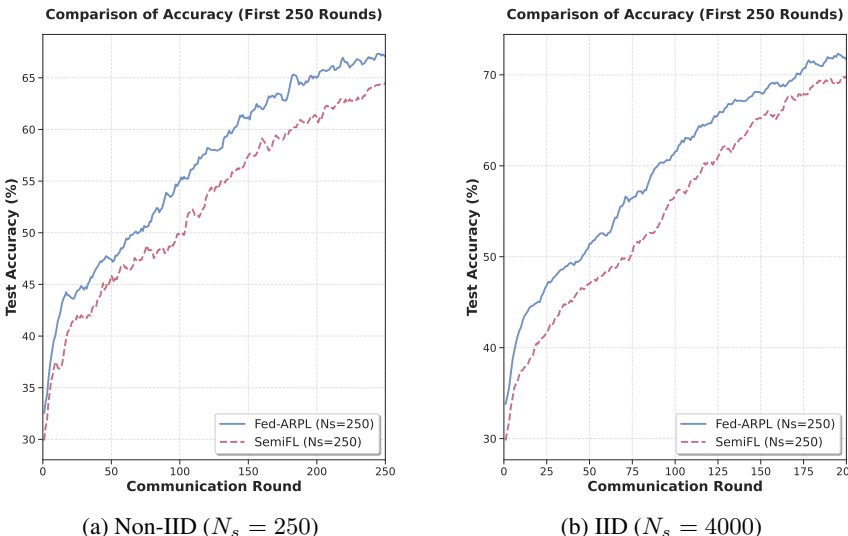

(a) Non-IID ($N_s = 250$)  (b) IID ($N_s = 4000$)

Figure 6: Comparison of convergence speed during the initial training phase on CIFAR-10. Fed-ARPL consistently demonstrates faster convergence than the SemiFL baseline in both (a) a challenging Non-IID setting and (b) a stable IID setting, effectively mitigating the "cold-start" problem.

### G.1  EFFECTIVENESS ON MITIGATING THE COLD-START PROBLEM

To substantiate our claim of overcoming the "cold-start" problem, we compare the initial learning curves of Fed-ARPL against the SemiFL baseline on CIFAR-10. As illustrated in Figure 6, Fed-ARPL consistently achieves faster convergence in both IID and the more challenging Non-IID settings. For instance, in the Non-IID scenario (Figure 6a), Fed-ARPL reaches $60\%$ accuracy around round 95, whereas the baseline requires approximately 115 rounds to achieve the same milestone, demonstrating a significant learning acceleration.

This rapid convergence stems from the synergistic effect of our first two phases. The **Adaptive Thresholding** mechanism immediately activates the vast unlabeled client data, breaking the initial inertia that plagues fixed-threshold methods. Concurrently, the **One-Way Prototypical Guidance** provides a stable, structural learning target that regularizes the training process against the noisy pseudo-labels of this nascent phase. Together, these mechanisms successfully transform the typically inert "cold-start" period into one of rapid and efficient guided learning.

### G.2  ABLATION STUDY ON ROBUSTNESS WITHOUT CONSISTENCY REGULARIZATION

To further investigate the robustness and generalizability of Fed-ARPL's core components, we conducted an additional ablation study. In this experiment, we deliberately replaced the standard consistency regularization loss (Eq. equation 1), which is based on strong data augmentation, with a more fundamental supervised cross-entropy loss applied directly to the pseudo-labeled data. This setup allows us to isolate and evaluate the effectiveness of our adaptive thresholding and prototypical learning mechanisms, independent of advanced SSL techniques like FixMatch. The experiment was performed using the ResNet-18 architecture on CIFAR-10 ($N_s = 250$, Non-IID $\alpha = 0.5$).

Table 4: Ablation study on the robustness of Fed-ARPL's components when the consistency regularization loss is replaced with a standard cross-entropy loss on pseudo-labels. The experiment is run on CIFAR-10 ($N_s = 250$, Non-IID $\alpha = 0.5$) with ResNet-18.

| Framework Configuration | Accuracy (%) |
|---|---|
| *(a)* Baseline (w/o Adaptive Threshold & Protos) | 64.53 |
| *(b)* + Adaptive Thresholding (Phase 1) | 65.75 |
| *(c)* + Teacher-Guided Proto. (Phase 2) | 68.02 |
| **(d) + Reciprocal Learning (Fed-ARPL)** | **69.45** |

The results, presented in Table 4, clearly show that the core mechanisms of Fed-ARPL remain highly effective. Starting from a baseline that uses only a fixed threshold and this basic supervised loss on pseudo-labels *(a)*, the introduction of our **Adaptive Thresholding** mechanism *(b)* provides a notable gain of $1.22\%$. Building upon this, the addition of **Teacher-Guided Prototyping** *(c)* further boosts the accuracy by a significant $2.27\%$. Finally, the complete Fed-ARPL framework with **Reciprocal Learning** *(d)* achieves the highest accuracy of $69.45\%$, adding another $1.43\%$. This demonstrates that our proposed adaptive and reciprocal learning strategies are not merely enhancements to a specific SSL loss, but are fundamental, independently effective mechanisms for improving FSSL. Their synergistic contributions provide significant performance gains, highlighting the robustness of the Fed-ARPL framework.

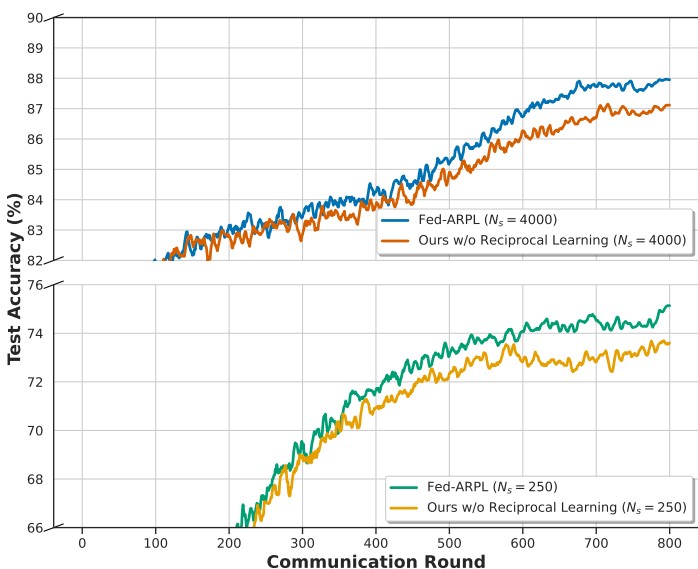

Figure 7: Full training trajectory on CIFAR-100 ($N_s = 2500$, IID). The performance gap between Fed-ARPL and SemiFL widens significantly in the later stages, demonstrating Fed-ARPL's ability to break the knowledge bottleneck.

### G.3 EFFECTIVENESS ON BREAKING THE KNOWLEDGE BOTTLENECK

The ultimate test of our framework is its ability to break the performance ceiling imposed by the server's limited data. To isolate the impact of our reciprocal learning phase, we compare the full Fed-ARPL framework ("FeedBack") against a variant that remains perpetually in Phase 2 ("NoFeedBack"). As shown in Figure 7, the "FeedBack" mechanism consistently yields superior performance. The performance gap is particularly pronounced in the low-label regime (CIFAR-10, $N_s = 250$), where the "FeedBack" curve steadily pulls away from the "NoFeedBack" curve, demonstrating that integrating client knowledge is critical for achieving a better solution. Even with more labeled data ($N_s = 4000$), the reciprocal learning phase still provides a consistent, albeit smaller, performance

advantage. This sustained improvement is a direct consequence of the Reciprocal Learning in Phase 3. By enabling proficient clients to contribute their high-quality local prototypes, Fed-ARPL infuses the global model with diverse insights from the vast unlabeled data, effectively shattering the knowledge bottleneck and converging to a superior global solution.

### G.4 ROBUSTNESS TO SERVER LABEL NOISE

To further evaluate the robustness of Fed-ARPL in realistic scenarios where server-side data might not be perfectly annotated, we conducted an additional series of experiments with symmetric label noise injected into the server's labeled dataset $\mathcal{D}_s$.

**Experimental Setup.** We selected the challenging CIFAR-100 dataset with $N_s = 2500$ labeled samples and a Non-IID partition of $\mathrm{Dir}(\alpha = 0.5)$. We introduced symmetric label noise at varying ratios: $0\%$ (clean baseline), $5\%$, $10\%$, and $20\%$. For a given noise ratio $r$, $r\%$ of the labels in $\mathcal{D}_s$ were randomly flipped to one of the other classes with uniform probability. We compare the performance of Fed-ARPL against the strong baseline, SemiFL.

**Results and Analysis.** The results are summarized in Table 5. As the noise level increases, the performance of both methods naturally degrades. However, Fed-ARPL demonstrates significantly higher resilience compared to SemiFL.

Table 5: Robustness comparison under varying levels of symmetric label noise on the server dataset. Experiments are conducted on CIFAR-100 ($N_s = 2500$, Non-IID $\alpha = 0.5$). We report the final test accuracy (%).

| Method | Label Noise Ratio | | | |
|---|---|---|---|---|
| | 0% (Clean) | 5% | 10% | 20% |
| SemiFL | 43.53 | 38.09 | 34.31 | 26.70 |
| **Fed-ARPL** | **50.97** | **47.53** | **43.76** | **35.36** |
| *Improvement* | *+7.44* | *+9.44* | *+9.45* | *+8.66* |

Under clean conditions ($0\%$ noise), Fed-ARPL already outperforms SemiFL by $7.44\%$. Crucially, as the noise ratio increases to $5\%$ and $10\%$, this performance gap widens to nearly $9.5\%$. Even under severe noise ($20\%$), where one in five server labels is incorrect, Fed-ARPL maintains a respectable accuracy of $35.36\%$, whereas SemiFL's performance collapses to $26.70\%$, a gap of $8.66\%$.

This superior robustness can be attributed to the prototypical guidance mechanism in Fed-ARPL. While standard cross-entropy loss (used in SemiFL) is notoriously sensitive to noisy labels, prototypes represent the geometric centroid of a class. This averaging effect inherently dampens the influence of individual mislabeled outliers. By guiding clients towards these more robust class centroids rather than forcing them to overfit to specific noisy samples, Fed-ARPL effectively regularizes the training process, preventing the propagation of server-side errors to the client models.

### G.5 EFFECTIVENESS UNDER EXTREME LABEL SCARCITY

To evaluate the limits of our framework, we conducted stress tests under conditions of extreme label scarcity, where the server holds fewer than 10 samples per class. Specifically, we tested on CIFAR-10 with $N_s = 50$ and CIFAR-100 with $N_s = 500$, both corresponding to a **5-shot per class** scenario.

**Experimental Setup.** For CIFAR-10, we used a Non-IID partition with $\mathrm{Dir}(\alpha = 0.5)$ and $N_s \in \{50, 100\}$. For CIFAR-100, we used an IID partition with $N_s \in \{500, 1000\}$. All other hyperparameters remained consistent with the main experiments.

**Results and Analysis.** The results are presented in Table 6. Fed-ARPL consistently outperforms the SemiFL baseline across all settings.

Table 6: Performance comparison under extreme label scarcity (5 and 10 samples/class). We report the final test accuracy (%).

| Dataset | Setting | Method | Labeled Samples ($N_s$) | |
|---|---|---|---|---|
| | | | 5 samples/cls | 10 samples/cls |
| CIFAR-10 (Dir 0.5) | $N_s \in \{50, 100\}$ | SemiFL | 45.89 | 55.47 |
| | | **Fed-ARPL** | **47.71** | **57.13** |
| | | *Improvement* | *+1.82* | *+1.66* |
| CIFAR-100 (IID) | $N_s \in \{500, 1000\}$ | SemiFL | 26.07 | 31.39 |
| | | **Fed-ARPL** | **27.14** | **32.16** |
| | | *Improvement* | *+1.07* | *+0.77* |

Even when the server provides minimal supervision (5 samples/class), Fed-ARPL maintains a performance advantage. On CIFAR-10 ($N_s = 50$), it achieves a gain of $1.82\%$. This confirms that our adaptive thresholding mechanism successfully activates client learning even when the initial teacher is weak, and that the prototypical guidance, though derived from sparse data, still offers a better regularization signal than a purely gradient-based update from a high-variance teacher.

### G.6 GENERALIZABILITY ON HIGHER-RESOLUTION DATA

To address concerns regarding the generalizability of our framework to datasets with higher image resolutions than standard CIFAR benchmarks ($32 \times 32$), we conducted additional experiments on the STL-10Coates et al. (2011) dataset. STL-10 contains images with a resolution of $96 \times 96$ pixels, representing a 9-fold increase in input dimensionality.

**Experimental Setup.** We evaluated Fed-ARPL and the SemiFL baseline under two data distribution settings: IID and Non-IID with Dir($\alpha = 0.5$). The server labeled set size was fixed at $N_s = 1000$. Due to the increased computational cost of higher-resolution training, both methods were trained for 200 communication rounds. All other hyperparameters followed the standard configuration used in our main experiments.

**Results and Analysis.** The results are summarized in Table 7. Fed-ARPL consistently outperforms SemiFL in both IID and Non-IID scenarios.

Table 7: Test accuracy (%) on the higher-resolution STL-10 dataset ($96 \times 96$) with $N_s = 1000$. Results are reported after 200 communication rounds.

| Method | IID | Non-IID (Dir 0.5) |
|---|---|---|
| SemiFL | 76.59 | 71.55 |
| **Fed-ARPL** | **77.01** | **72.10** |
| *Improvement* | *+0.42* | *+0.55* |

Despite the limited training rounds, Fed-ARPL establishes a clear performance advantage. In the Non-IID setting, the gap widens to $0.55\%$, suggesting that our prototype-based reciprocal learning is particularly effective at handling statistical heterogeneity even in higher-dimensional feature spaces. These findings confirm that the benefits of our framework extend beyond small-scale benchmarks and are applicable to more complex, real-world visual tasks.

