# OpenReview forum: "Fed-ARPL: Adaptive and Reciprocal Prototype Learning for Semi-supervised Federated Learning"
_ICLR.cc/2026/Conference — ICLR 2026 Conference Withdrawn Submission_

### Official Review · Reviewer_TMcw · 2025-10-29

**Soundness:** 3
**Presentation:** 3
**Contribution:** 2
**Rating:** 4
**Confidence:** 5

**Summary:**

This paper proposes Fed-ARPL, a comprehensive framework for Federated Semi-Supervised Learning (FSSL) that aims to overcome two major challenges — the Cold-start problem and the Knowledge Bottleneck.
Fed-ARPL adopts a three-phase adaptive learning strategy: an adaptive warm-up phase that dynamically selects pseudo-labeled data, a teacher-guided phase where the server provides global prototype supervision, and a student-feedback phase where clients contribute local prototypes back to the server.
Extensive experiments on CIFAR-10, CIFAR-100, and SVHN under various non-IID and label-scarce conditions show that Fed-ARPL achieves substantial gains over existing baselines, validating the effectiveness of its design. Ablation studies further validate the contribution of each proposed component.

**Strengths:**

1. The paper introduces a complete and coherent framework, Fed-ARPL, that systematically tackles both the Cold-start and Knowledge Bottleneck challenges in FSSL.

2. Experimental results across multiple datasets and varying heterogeneity levels clearly demonstrate that Fed-ARPL significantly outperforms existing baselines. The improvements are especially notable under severe label scarcity and strong non-IID conditions, underscoring the method’s robustness and generalizability.

3. The paper is clearly written, well-structured, and easy to follow.

**Weaknesses:**

1. While the integration of adaptive thresholding and prototype feedback into a three-phase pipeline is well-engineered, each individual component (e.g., prototypical aggregation) builds upon existing concepts. The novelty lies more in the combination and systematization than in a fundamentally new learning principle.

2.  The framework introduces many additional hyperparameters (e.g., τ_fss, σ_fss, λ_plc, W_gcplca, τ′, α_t, β) that must be manually tuned. This reduces practicality, as many of these thresholds require non-trivial calibration and could vary widely across datasets.

3. The baselines are relatively dated; the newest comparison (FL2, 2024) is over a year old. The paper would be more convincing if compared with very recent FSSL frameworks such as FedLGMatch or other adaptive pseudo-labeling methods from 2025.

**Questions:**

1. Could the authors explore adaptive or self-tuning mechanisms for the many hyperparameters, especially τ_fss, σ_fss, and W_gcplca? These parameters could potentially be inferred based on model confidence dynamics rather than fixed thresholds.

2.  What would happen if the server-side labeled data were even more limited—say, fewer than 10 samples per class? Would the adaptive thresholding still produce stable pseudo-labels, or would the teacher-guided phase fail to provide useful supervision?

---

> ### Author Response · Authors · 2025-11-28
> **W1: Novelty Concerns (Combination vs. Synergistic Co-design). W2 & Q1: Hyperparameter Sensitivity and Self-Tuning Mechanisms.**
>
> **W1: Novelty Concerns (Combination vs. Synergistic Co-design).**
>
> **Response:**
>
> We respectfully disagree with the view that our novelty is limited to "combination." While the individual components (adaptive thresholds, prototypes) have precedents, our core contribution lies in (1) the synergistic co-design of these mechanisms to solve a specific sequence of FSSL challenges, and (2) the fundamental redefinition of the prototype's role within this pipeline.
>
> **First**, our framework is not a simple concatenation of existing modules. The mechanisms in our early phases are critically interdependent: while Adaptive Thresholding successfully activates dormant unlabeled data (solving the Cold-Start), it inevitably introduces significant label noise. The One-Way Prototypical Guidance is specifically introduced not just as a standard regularizer, but as a necessary countermeasure to this noise, providing a stable structural anchor. This specific synergy—transforming the "cold-start" into a phase of *rapid yet disciplined* convergence—is a novel architectural contribution that simple combination cannot achieve.
>
> **Second**, and more importantly, we introduce a dynamic role evolution for prototypes that fundamentally differs from prior arts. In existing works (e.g., FedProto), prototypes serve primarily as static, top-down regularizers. In contrast, Fed-ARPL orchestrates an evolution: prototypes begin as *teacher-guided anchors* in Phase 2 to stabilize learning, but crucially evolve into *vehicles for student feedback* in Phase 3. This paradigm shift—from unidirectional regulation to bi-directional knowledge co-creation—is the key innovation that allows our framework to break the "Knowledge Bottleneck."
>
> **Action:** We have revised the **Introduction** and **Related Work** to explicitly articulate this co-design philosophy and the novel, evolving nature of our prototype mechanism.
>
> ---
>
> **W2 & Q1: Hyperparameter Sensitivity and Self-Tuning Mechanisms.**
>
> **Response:**
>
> We thank the reviewer for raising this important practicality concern. While the list of hyperparameters appears extensive, our empirical experience suggests they are robust and do not require complex per-dataset calibration.
>
> **First**, regarding Practicality and Sensitivity, we would like to highlight that we used a unified set of hyperparameters across all our experiments (CIFAR-10, CIFAR-100, SVHN) and all heterogeneity settings. For instance, the transition windows ($W=5$) and stability thresholds ($σ=0.3/0.05$) were fixed and not tuned individually for each case. The fact that Fed-ARPL consistently achieves SOTA performance across such diverse scenarios with a fixed configuration strongly suggests that these hyperparameters are not brittle and the method generalizes well without extensive manual tuning.
>
> **Second**, regarding Self-Tuning Mechanisms, we fully agree with the reviewer's insightful suggestion. Exploring adaptive criteria is a logical next step.
>
> - **Feasibility:** Instead of fixed thresholds (e.g., $τ_{fss}$), one could monitor the rate of change (derivative) of the FSS or GCPLCA metrics. A transition could be triggered when the metric enters a plateau (i.e., gradient≈0 ), signaling that the current phase has yielded its maximum benefit.
> - **Future Direction:** While developing and validating a robust meta-learning or heuristic-based auto-tuning module is beyond the scope of this current work, we have explicitly acknowledged this as a promising avenue to further enhance the framework's "plug-and-play" nature.
>
> **Action:** We have added a new paragraph titled **"Robustness and Unified Configuration"** in **Appendix C.3**, explicitly clarifying that a fixed set of hyperparameters was used across all datasets to demonstrate robustness. We also included the discussion on potential self-adaptive mechanisms (based on metric dynamics) as a future direction in the same section.

---

> ### Author Response · Authors · 2025-11-28
> **W3: Baseline Recency (e.g., FedLGMatch). Q2: Performance under Extreme Label Scarcity (≤10 samples/class).**
>
> **W3: Baseline Recency (e.g., FedLGMatch).**
>
> **Response:**
>
> We appreciate the reviewer for pointing us to the very latest advancements in the field. We agree that comparing against 2025 frameworks would further strengthen our evaluation.
>
> However, regarding FedLGMatch and other 2025 adaptive methods, we faced a practical constraint: at the time of our submission and current rebuttal, no official source code or reproducible implementation was publicly available.
>
> - **Fairness Concern:** Implementing complex SOTA methods from scratch without official reference carries a high risk of misconfiguration, which could lead to an unfair underestimation of their true performance. To ensure the integrity of our empirical evaluation, we restricted our direct quantitative comparison to strong, open-source baselines (e.g., SemiFL, (FL)$^2$, pFedKnow) where we could guarantee a rigorous "apples-to-apples" setup.
> - **Qualitative Comparison:** Nevertheless, we recognize the conceptual relevance of these works. We have updated our Related Work section to discuss FedLGMatch, highlighting how our reciprocal prototype evolution differs from their graph-based matching approach. We are eager to include a quantitative comparison in the camera-ready version should the code become available.
>
> **Action:** We have updated the **Related Work** section to include a discussion on recent 2025 frameworks like FedLGMatch, contextualizing our contribution within this emerging landscape.
>
> ---
>
> **Q2: Performance under Extreme Label Scarcity (≤10 samples/class).**
>
> **Response:**
>
> This is an excellent stress test. To empirically answer this, we conducted new experiments under two levels of extreme scarcity: 5 samples/class（$N_s=50/500$) and 10 samples/class ($N_s=100/1000$).
>
> This is an excellent stress test for our adaptive mechanisms. To empirically answer this, we conducted **new experiments** under extreme scarcity:
>
> 1. **CIFAR-10:**  Non-IID (Dir 0.5) with $N_s=50$ and $100$.
> 2. **CIFAR-100:** IID with $N_s=500$ and $1000$.
>
> **Results:** As detailed in the new Appendix G, Fed-ARPL demonstrates remarkable resilience even in these "few-shot" server scenarios.
>
> - CIFAR-10 (Dir 0.5):
>     - 5-shot ($N_s=50$): Fed-ARPL achieves 47.71%, outperforming SemiFL (45.89%) by 1.82%.
>     - 10-shot ($N_s=100$): Fed-ARPL achieves 57.13%, surpassing SemiFL (55.47%) by 1.66%.
> - CIFAR-100 (IID):
>     - 5-shot ($N_s=500$): Fed-ARPL achieves 27.14% vs. SemiFL's 26.07% (+1.07%).
>     - 10-shot ($N_s=1000$): Fed-ARPL achieves 32.16% vs. SemiFL's 31.39% (+0.77%).
>
> Why it works: Under such scarcity, the initial teacher model is indeed weak. However, our Adaptive Thresholding automatically lowers the confidence bar significantly in the early rounds, preventing the training from stalling (which happens with a fixed $τ=0.95$). Crucially, while the initial prototypes are noisy, they still provide a better-than-random structural prior compared to the high-variance gradients of a few-shot model, effectively guiding the clients until they can provide reciprocal feedback.
>
> **Action:** We have added these results to **Appendix G**, demonstrating that our framework maintains its advantage even when server supervision is critically limited.

---

> ### Author Response · Authors · 2025-11-28
> **We are looking forward to your feedback.**
>
> Dear Reviewer TMcw,
>
> We extend our sincere gratitude for your high-confidence and expert review. Your insightful critique regarding the architectural novelty, hyperparameter practicality, and evaluation scope has driven us to significantly elevate the rigor of our work.
>
> Specifically, your comments motivated us to: (1) articulate the unique "synergistic co-design" nature of our framework more clearly; (2) conduct critical stress tests under extreme label scarcity (down to 5 samples/class); and (3) expand our theoretical and practical discussions on hyperparameter robustness and privacy.
>
> We believe these substantial additions, particularly the new experimental evidence in Appendix G and the clarified positioning against recent works, directly address your concerns. We hope this enhanced manuscript demonstrates the true value of Fed-ARPL, and we look forward to your re-evaluation.
>
> Sincerely,
>
> Authors of Submission 10229

---

### Official Review · Reviewer_uo4K · 2025-10-29

**Soundness:** 3
**Presentation:** 3
**Contribution:** 2
**Rating:** 2
**Confidence:** 4

**Summary:**

The paper addresses the "labels-at-server" Federated Semi-Supervised Learning (FSSL) setting, where a central server holds limited labeled data and clients have only unlabeled samples. It identifies two key challenges: the cold-start problem caused by fixed pseudo-label thresholds, and a knowledge bottleneck from limited server supervision. The proposed Fed-ARPL framework with phase transitions governed in a formulated manner.

**Strengths:**

The paper provides a clear decomposition of the training dynamics. Its three-phase design distinctly separates early data utilization, representation manifold shaping, and capacity expansion. The phase-transition criteria are formulated, making the pipeline more principled than heuristic. Experimental results are competitive with the evolution of FSS/GCPLCA metrics and threshold acceptance rates that enhances interpretability.

**Weaknesses:**

While the paper is well structured, its novelty over prior FSSL and FL-prototype lines is incremental. Adaptive or confidence-based thresholds and prototype sharing are established ideas. The main contribution appears to lie in their orchestration via the proposed gating metrics. The evaluation scope is limited to small-scale, low-resolution datasets, and lacks robustness tests under server label noise, client drift, or open-set unlabeled pools. Privacy and communication trade-offs are underexplored (prototype uploads may leak distributional information, and their cost relative to model-only is not quantified). The phase criteria (FSS, GCPLCA) are empirically tuned without theoretical grounding while sensitivity analyses are dataset-specific. Finally, several presentation issues remain (e.g., typos (e.g., “Fed-APRL”), unresolved equation references (e.g., "??"), inconsistent notation).

**Questions:**

N/A

---

> ### Author Response · Authors · 2025-11-28
> **W1: Novelty Concerns (Orchestration vs. Synergistic Co-design). W2: Evaluation Scope and Robustness Tests.**
>
> **W1: Novelty Concerns (Orchestration vs. Synergistic Co-design).**
>
> **Response:**
>
> We respectfully disagree with the characterization of our novelty as merely incremental "orchestration." While adaptive thresholds and prototype sharing are indeed established concepts in isolation, our core contribution lies in the synergistic co-design of these mechanisms to solve a specific sequence of FSSL challenges, and the fundamental redefinition of the prototype's role.
>
> **First**, our framework is not a simple concatenation of existing modules. The mechanisms in our early phases are critically interdependent: while Adaptive Thresholding successfully activates dormant unlabeled data, it inevitably introduces significant label noise. The One-Way Prototypical Guidance is specifically introduced not just as a standard regularizer, but as a necessary countermeasure to this noise, providing a stable structural anchor. This specific synergy—transforming the "cold-start" into a phase of *rapid yet disciplined* convergence—is a novel architectural contribution that prior works lack.
>
> **Second**, and more importantly, we introduce a dynamic role evolution for prototypes that fundamentally differs from prior arts like FedProto. In existing works, prototypes serve primarily as static, top-down regularizers for heterogeneity. In contrast, Fed-ARPL orchestrates an evolution: prototypes begin as *teacher-guided anchors* in Phase 2 to stabilize learning, but crucially evolve into *vehicles for student feedback* in Phase 3. This paradigm shift—from unidirectional regulation to bi-directional knowledge co-creation—is the key innovation that allows our framework to break the knowledge bottleneck, a capability unattainable by simple orchestration.
>
> **Action:** We have revised the **Introduction** and **Related Work** to explicitly articulate this co-design philosophy and the novel, evolving nature of our prototype mechanism.
>
> **W2: Evaluation Scope and Robustness Tests.**
>
> **Response:**
>
> We thank the reviewer for highlighting the importance of rigorous stress testing. We acknowledge that validating our framework on more challenging scenarios is essential to demonstrate its robustness and generalizability.
>
> **First**, regarding the robustness against noise, we have conducted comprehensive additional experiments to test Fed-ARPL's resilience under server-side label noise, a critical factor in real-world deployments. We introduced symmetric label noise at varying levels ($0\%, 5\%, 10\%, 20\%$) into the server's dataset on CIFAR-100 ($N_s=2500$, Non-IID $α=0.5$). As detailed in the new Appendix G, Fed-ARPL demonstrates superior resilience compared to the SemiFL baseline. Notably, under severe noise ($20\%$), Fed-ARPL maintains an accuracy of 35.36%, significantly outperforming SemiFL (26.70%) by a margin of 8.66%. This result confirms that our prototype-based guidance effectively regularizes the model against fitting to noisy supervision, preventing the performance collapse observed in baselines.
>
> **Second,** concerning the evaluation scope, we would like to clarify that our chosen datasets (CIFAR-10/100, SVHN) and settings align with the standard benchmarks in seminal "labels-at-server" FSSL literature [e.g., SemiFL, (FL)²]. However, to further address the concern about "low-resolution," we have included results on STL-10 (96x96 resolution, 9x larger than CIFAR). As detailed in Appendix G, Fed-ARPL achieves 77.01% (IID) and 72.10% (Non-IID) accuracy on STL-10, consistently outperforming the SemiFL baseline (76.59% and 71.55%). This verifies our method's efficacy on higher-resolution, complex visual tasks.
>
> **Action:** We have added a new section in **Appendix G** containing both the "Robustness to Server Label Noise" analysis and the "Generalizability on Higher-Resolution Data" (STL-10) results to broaden the evaluation scope beyond standard benchmarks.

---

> ### Author Response · Authors · 2025-11-28
> **W3: Privacy Risks and Communication Trade-offs. W4: Theoretical Grounding of Phase Criteria and Presentation Issues.**
>
> **W3: Privacy Risks and Communication Trade-offs.**
>
> **Response:**
>
> We appreciate the reviewer for raising these critical practical considerations. We agree that a comprehensive evaluation must transparently account for the system overheads and potential privacy implications introduced by prototype exchange.
>
> **First**, regarding communication overhead, we have conducted a rigorous quantitative analysis to demonstrate that the cost of transmitting prototypes is negligible. For a standard WideResNet-28x2 model (∼5.88MB), uploading 100 class prototypes (each 128-dim, total∼51KB) increases the payload by only 0.87%, even in the most communication-intensive reciprocal phase. This marginal increase in bandwidth is a highly favorable trade-off given the significant performance gains (e.g., +4.11% on CIFAR-100) achieved by our method.
>
> **Second**, regarding privacy risks, we acknowledge that prototype uploads could theoretically leak distributional information. However, it is important to note that prototypes are aggregated statistics (mean embeddings) computed on pseudo-labeled data. This aggregation, combined with the inherent noise of pseudo-labels, provides a natural layer of data obfuscation. Furthermore, our framework is fully compatible with advanced privacy-enhancing technologies. As a future direction, we propose integrating Secure Aggregation or applying Differential Privacy to the prototypes, which would mathematically guarantee that the server only receives the global consensus without accessing individual client distributions.
>
> **Action:** We have added a new **Appendix F** ("System Overhead and Privacy Analysis") to provide the detailed communication cost breakdown and a comprehensive discussion on privacy risks and potential mitigations.
>
> ---
>
> **W4: Theoretical Grounding of Phase Criteria and Presentation Issues.**
>
> **Response:**
>
> We thank the reviewer for this insightful comment. We agree that grounding empirical heuristics in established theory significantly strengthens the methodology. We also apologize for the presentation oversights.
>
> **First**, regarding the theoretical justification, we have added a comprehensive discussion in Appendix E to clarify that our metrics are not arbitrary. FSS (Feature-Space Separation) is theoretically rooted in Deep Metric Learning principles (e.g., Fisher Discriminant Analysis), aiming to maximize inter-class separation while minimizing intra-class variance. GCPLCA serves as a direct proxy for the system's Epistemic Uncertainty. A high and stable GCPLCA indicates that the client collective has converged to a low-entropy state, signaling that their knowledge is reliable enough for reciprocal feedback. We have also explicitly positioned the formal convergence analysis of these dynamic criteria as a key direction for future work in the Conclusion.
>
> **Second**, regarding presentation, we have conducted a meticulous proofreading of the entire manuscript. We have corrected all identified typos (e.g., "Fed-ARPL"), resolved all latent equation references (e.g., Eq. ??), and ensured consistent notation across all sections to meet the high standards of the venue.
>
> **Action:** We have added **Appendix E** ("On the Design of Phase Transition Metrics") to provide the theoretical grounding and have submitted a revised manuscript with all presentation errors corrected.

---

> ### Author Response · Authors · 2025-11-28
> **We are looking forward to your feedback.**
>
> Dear Reviewer uo4K,
>
> We express our sincere gratitude for your detailed and rigorous review. Your constructive skepticism regarding the novelty and robustness of our work has pushed us to significantly strengthen our manuscript. Specifically, your comments motivated us to conduct critical stress tests under label noise, expand our theoretical justifications, and more clearly articulate the synergistic nature of our framework.
>
> We believe that these substantial additions—including the new robustness experiments in Appendix G and the theoretical grounding in Appendix D—have directly addressed your core concerns. We hope that our clarifications and the enhanced evidence presented in the revised paper will persuade you to reconsider your assessment of our contribution.
>
> Thank you once again for your time and for helping us improve the quality of this work.
>
> Sincerely,
>
> Authors of Submission 10229

---

### Official Review · Reviewer_9v6v · 2025-10-30

**Soundness:** 2
**Presentation:** 3
**Contribution:** 2
**Rating:** 4
**Confidence:** 3

**Summary:**

This paper proposes Fed-ARPL, an Adaptive and Reciprocal Prototype Learning framework for federated semi-supervised learning. It focuses on two key and relatively understudied challenges: the cold-start problem and the knowledge bottleneck. To mitigate these problems, a three-phase training strategy is proposed: an adaptive thresholding mechanism to enhance early pseudo-label utilization, a teacher-guided prototypical learning phase to ensure global feature consistency, and a student-feedback phase enabling bidirectional knowledge exchange between clients and server. Experiments on CIFAR-10, CIFAR-100, and SVHN demonstrate that Fed-ARPL achieves state-of-the-art performance and robust convergence under both IID and Non-IID data distributions.

**Strengths:**

1.	The motivation of this work is convincing. It targets two fundamental and practical challenges in FSSL—the cold-start and knowledge bottleneck problems.
2.	Experimental results on CIFAR-10, CIFAR-100, and SVHN under multiple Dirichlet skews and IID settings show consistent and significant performance gains.

**Weaknesses:**

1.	Fed-ARPL requires each client to upload both model weights and class prototypes to the server, whereas many related works transmit only one of these components. This design choice may raise two questions: (a) Communication and computation: Does the inclusion of both significantly increase communication or computational cost compared to existing methods? A brief quantitative analysis would help clarify this. (b) Data privacy: When the server has access to both local model parameters and class prototypes, could this combination potentially increase the risk of information leakage or client data inference? It would be valuable if the authors could provide a deeper discussion or empirical assessment regarding privacy implications in this context.
2.	This work evaluates the proposed method on three widely used but relatively small image datasets (CIFAR-10, CIFAR-100, and SVHN), all with 32×32 image resolution. Would the proposed framework maintain its advantages on larger and more complex datasets (e.g., ImageNet)? Including at least one large-scale experiment could strengthen the generalizability of the proposed method. In addition, several related methods, like FedProto, are discussed in the related work section but not included in the experimental comparison. Could the authors provide some justification for the chosen baselines?
3.	Lack of theoretical justification for the new metrics (FSS & GCPLCA). The paper provides useful empirical and sensitivity analyses, but the motivation for selecting these specific metrics over standard alternatives (e.g., Silhouette or Davies-Bouldin for feature separability) could be discussed in more detail to improve clarity.
4.	There are a few missing references, like Eq.equation??.

**Questions:**

The questions have been stated in the weakness section. I would raise my score if these questions could be properly addressed in the rebuttal.

---

> ### Author Response · Authors · 2025-11-28
> **W1 : System Overhead (Communication/Computation) and Privacy Implications.**
>
> **Response:**
>
> We thank the reviewer for these critical questions regarding the practical trade-offs of our design. We agree that introducing prototype exchange warrants a careful examination of both efficiency and privacy.
>
> **First**, regarding Communication and Computation Cost, we have conducted a rigorous quantitative analysis to demonstrate that the overhead is negligible.
>
> - **Communication:** Prototypes are extremely compact. For a standard WideResNet-28x2 backbone ($D=128$), uploading 100 class prototypes consumes only ∼51KB. In contrast, the model parameters themselves are 5.88MB. Consequently, even in the reciprocal phase where both global and local prototypes are exchanged, the total communication payload increases by only ∼0.87% compared to a model-only baseline. This marginal cost yields significant accuracy gains (e.g., +4.11% on CIFAR-100).
> - **Computation:** The calculation of local prototypes requires only a single forward pass over a data subset and a simple averaging operation. Compared to the computationally intensive local training loop (5 epochs of backpropagation), this step adds less than 0.5% to the client's computation time. Thus, Fed-ARPL imposes virtually no additional latency compared to existing methods.
>
> **Second**, regarding Data Privacy, we acknowledge that simultaneous access to model updates and prototypes could theoretically increase information leakage risks. However, two factors mitigate this concern in our context:
>
> 1. **Aggregation as Obfuscation:** Prototypes are aggregated statistics (mean embeddings) computed on pseudo-labeled data. This averaging process, combined with the inherent noise of pseudo-labels in semi-supervised learning, provides a natural layer of data obfuscation that makes reconstructing individual raw samples difficult.
> 2. **Compatibility with PETs:** Our framework is fully compatible with advanced Privacy-Enhancing Technologies (PETs). As a concrete mitigation strategy, we propose integrating Secure Aggregation or applying Differential Privacy to the prototype vectors. These techniques would mathematically guarantee that the server receives only the global consensus without accessing individual client distributions, effectively neutralizing the inference risk.
>
> **Action:** We have added a new **Appendix F** ("System Overhead and Privacy Analysis") to provide the detailed mathematical breakdown of costs and a comprehensive discussion on privacy risks and potential mitigations as a key direction for future work.

---

> ### Author Response · Authors · 2025-11-28
> **W2: Generalizability on Larger Datasets and Baseline Selection Rationale.**
>
> **Response:**
>
> We appreciate the reviewer's suggestion to strengthen our evaluation scope. We acknowledge that validating performance on higher-resolution, complex datasets is essential for demonstrating generalizability.
>
> **First**, regarding Large-Scale Experiments, to verify our method's efficacy on higher-resolution images without incurring the prohibitive computational cost of full-scale ImageNet training, we have added experiments on STL-10. This dataset features 96x96 resolution images (9x larger than CIFAR) sourced from ImageNet, presenting a significantly more complex visual recognition challenge.
>
> - **Results:** As shown in the new Table 7 (Appendix G), Fed-ARPL demonstrates consistent superiority on STL-10, ). In the IID setting, it achieves 77.01% accuracy, outperforming the SemiFL baseline (76.59%). In the more challenging Non-IID setting, it maintains its lead with 72.10% against 71.55%. These results confirm that our prototype-based guidance remains effective and robust even when dealing with high-dimensional feature spaces and complex real-world objects.
>
> **Second**, regarding Baseline Selection (FedProto), we carefully selected baselines (SemiFL, FedMatch, (FL)$^2$, pFedKnow) that are specifically tailored for the "labels-at-server" FSSL setting.
>
> - **Task Mismatch:** While FedProto is a seminal work, it is originally designed for Supervised FL where clients possess fully labeled data. It lacks the intrinsic mechanisms (e.g., pseudo-labeling, consistency regularization) required to handle the unlabeled client data in our FSSL scenario. Directly comparing against it would require significant modifications, rendering the comparison unfair.
> - **Implicit Comparison:** However, our Phase 2 (Teacher-Guided Learning) can be viewed as an adaptation of the FedProto philosophy to the FSSL context. Our Ablation Study (Table 2) explicitly validates the contribution of this phase, thereby implicitly confirming the value of a FedProto-like mechanism within our framework.
>
> **Action:** We have clarified our baseline selection criteria in **Section 5** and added the STL-10 experimental results to **Appendix G** to demonstrate scalability and generalizability.

---

> ### Author Response · Authors · 2025-11-28
> **W3 & Q4: Theoretical Justification of Metrics and Presentation Corrections.**
>
> **Response:**
>
> We thank the reviewer for this insightful comment. We agree that grounding our empirical metrics in established theory and justifying our design choices against standard alternatives strengthens the paper's foundation. We also apologize for the presentation oversights.
>
> **First**, regarding the Motivation for Specific Metrics over Alternatives:
>
> - **Why not Silhouette/Davies-Bouldin?** While standard clustering indices like the Silhouette score are theoretically sound, they typically require computing pairwise distances between all samples, leading to a computational complexity of $O(N^2)$. In a federated setting with large local datasets, this is prohibitively expensive. In contrast, our FSS is explicitly designed to be prototype-centric, scaling linearly with the number of classes ($O(K^2)$). This not only ensures computational efficiency but also aligns perfectly with our framework's core mechanism, which operates on prototypes rather than raw sample clusters.
> - **Theoretical Grounding:** We have grounded FSS in Deep Metric Learning principles (maximizing inter-class separability vs. minimizing intra-class variance). Similarly, GCPLCA serves as a direct proxy for the system's Epistemic Uncertainty. A high and stable GCPLCA indicates that the client collective has converged to a low-entropy state, signaling that their knowledge is reliable enough for reciprocal feedback.
>
> **Second**, regarding Presentation, we have conducted a meticulous proofreading of the entire manuscript. We have corrected all identified typos and resolved all missing equation references (e.g., Eq. ??) to ensure the final version meets the highest standards.
>
> **Action:** We have added a new section in **Appendix E** ("On the Design of Phase Transition Metrics") to detail these theoretical justifications and comparisons. We have also corrected all presentation errors in the revised manuscript.

---

> ### Author Response · Authors · 2025-11-28
> **We are looking forward to your feedback.**
>
> Dear Reviewer 9v6v,
>
> We express our sincere gratitude for your detailed review. Your comments regarding system overhead, privacy, and metric justification have driven us to significantly enhance the rigor and completeness of our work.
>
> We believe that our quantitative cost analysis (showing <1% overhead), the new STL-10 experiments, and the expanded theoretical discussions in the Appendix directly address your concerns. We hope these clarifications and the revised manuscript demonstrate the soundness and contribution of Fed-ARPL, and we look forward to your re-evaluation.
>
> Sincerely,
>
> Authors of Submission 10229.

---

### Official Review · Reviewer_PQVk · 2025-10-30

**Soundness:** 2
**Presentation:** 2
**Contribution:** 3
**Rating:** 4
**Confidence:** 3

**Summary:**

Targets FSSL’s Cold-start (too-strict early thresholds block unlabeled use) and Knowledge Bottleneck (server’s small labeled set caps model). Proposes Fed-ARPL, a three-phase scheme: Warm-up with adaptive thresholding; Teacher-Guided with server prototypes; Student-Feedback where high-performing clients contribute refined prototypes back to the server. Shows SOTA on several benchmarks with ablations.

**Strengths:**

- Three-phase adaptive + prototypical strategy with reciprocal feedback.
- Addresses persistent pains (cold-start, bottleneck).
- Benchmarks and ablations are indicated.

**Weaknesses:**

- Missing specifics on when/how to switch phases; risk of premature or delayed transitions.
- Prototype quality under client drift/non-IID not rigorously assessed; could amplify bias.
- Absent systems accounting (communication, latency) for extra prototype exchanges.

**Questions:**

- What automated metrics trigger phase transitions and how robust are they?
- How do you filter noisy client prototypes to prevent degrading the global model?
- Please include compute/communication overhead vs. SemiFL/FL2 with identical hardware and rounds.

---

> ### Author Response · Authors · 2025-11-28
> **W1 & Q1: Specifics, Robustness, and Risks of Automated Phase Transitions.**
>
> **Response:**
>
> Thank you for this crucial question regarding the operational details and reliability of our phase transition mechanism. We acknowledge that the initial manuscript could have been more explicit about the safeguards against premature or delayed transitions.
>
> **First**, regarding the specifics of the trigger mechanism, our transitions are not governed by instantaneous metric values, which can be volatile in federated settings. Instead, we employ a strict dual-criterion stability check. A transition is triggered only when the monitored metric (FSS for Phase 2, GCPLCA for Phase 3) satisfies two conditions simultaneously over a predefined sliding window $W$:
>
> 1. **Sufficiency:** The moving average of the metric must exceed a high threshold $τ$, ensuring the model has accumulated sufficient structural quality (FSS) or confidence (GCPLCA).
> 2. **Stability:** Crucially, the standard deviation within the window must fall below a strict limit $σ$. This condition acts as a "damper," preventing premature transitions caused by transient spikes or unstable training phases.
> 3. **Sensitivity:** Our empirical results (e.g., consistent gains across diverse datasets/settings) implicitly demonstrate the robustness of these criteria. We have also clarified the specific hyperparameter values used in the Implementation Details.
>
> **Second**, regarding the **r**obustness and theoretical grounding, our metrics are not arbitrary heuristics. As we have now detailed in the new Appendix E, FSS is theoretically grounded in deep metric learning objectives, quantifying the geometric quality of the feature manifold (inter-class separation vs. intra-class compactness). GCPLCA serves as a proxy for the system's epistemic uncertainty; a high and stable value indicates that the client collective has converged to a low-entropy state where their knowledge is reliable enough for reciprocal feedback.
>
> **Finally**, empirically, we observe that this mechanism is highly robust. Across diverse datasets (CIFAR-10/100, SVHN) and heterogeneity levels (Dir $α∈{0.2,0.5,0.8}$), the same logic consistently identifies the appropriate transition points without requiring dataset-specific tuning of the window size, effectively mitigating the risk of suboptimal phase switching.
>
> **Action:** We have revised **Section 3.2** to explicitly describe this dual-criterion window mechanism and added **Appendix E** to provide the theoretical justification for the chosen metrics.

---

> ### Author Response · Authors · 2025-11-28
> **W2 & Q2: Prototype Quality Control and Noise Filtering under Non-IID.**
>
> **Response:**
>
> We appreciate the reviewer for raising this critical point. Ensuring the quality of client prototypes, especially under severe Non-IID conditions where local data may be biased, is indeed paramount to preventing the degradation of the global model. To address this, Fed-ARPL incorporates a systematic triple-safeguard mechanism designed to rigorously filter noise and mitigate bias amplification.
>
> **First**, we implement strict source filtering at the client side. In Phase 3, clients do not use all their data to compute prototypes. Instead, they apply a stricter confidence threshold $τ^′>τ^t$ to select only the highest-quality pseudo-labeled samples. This ensures that the uploaded prototypes are constructed exclusively from reliable, low-ambiguity data representations, effectively filtering out noise induced by local decision boundary uncertainty.
>
> **Second**, we employ weighted aggregation at the server side to handle statistical heterogeneity. As defined in Eq. (5), client prototypes are aggregated via a weighted average based on the number of high-confidence samples supporting each class. This mechanism naturally down-weights the contributions from clients who have sparse or unreliable data for a specific class—a common occurrence in Non-IID settings—thereby preventing minority outliers from skewing the global consensus.
>
> **Third**, and perhaps most importantly, we utilize teacher anchoring to stabilize the global update. In the reciprocal update step (Eq. (7)), we retain the server's own prototype $p_c^s$ as a stabilizing anchor. The adaptive weight $α_t$ ensures that the global consensus integrates client feedback gradually rather than drifting abruptly towards potentially biased client distributions.
>
> **Empirically**, the effectiveness of these safeguards is strongly supported by our results in Table 1. Under the extreme heterogeneity of Dir($α=0.2$), Fed-ARPL achieve 69.11%accuracy on CIFAR-10 ($N_s=250$), significantly outperforming the SemiFL baseline (67.59%). This demonstrates that our framework successfully extracts valuable knowledge from diverse clients without succumbing to the performance collapse often associated with unchecked prototype drift.
>
> **Action:** We have revised **Section 3.2.3** to explicitly discuss these three filtering and stabilizing mechanisms, highlighting their role in ensuring robustness against Non-IID shifts.

---

> ### Author Response · Authors · 2025-11-28
> **W3 & Q3: System Overhead (Communication & Computation) Analysis.**
>
> **Response:**
>
> We thank the reviewer for highlighting the importance of practical system constraints. We agree that a comprehensive evaluation must account for the additional costs introduced by prototype exchange. To this end, we have conducted a rigorous quantitative analysis comparing Fed-ARPL against standard baselines (e.g., SemiFL) under identical settings.
>
> **First**, regarding communication overhead, our analysis reveals that the cost of transmitting prototypes is negligible. Prototypes are compact vectors (e.g.,$100 classes×128 dim$ for WideResNet-28x2 on CIFAR-100), occupying only ∼51KB. In contrast, the model parameters themselves are significantly larger (∼5.88MB). Consequently, even in the reciprocal phase where both global and local prototypes are exchanged, the total communication payload increases by only 0.87% compared to a model-only baseline. This minimal bandwidth increase is a highly favorable trade-off for the substantial accuracy gains observed.
>
> **Second**, concerning computational latency, the additional burden on clients is virtually imperceptible. The computation of local prototypes requires only a single forward pass over a subset of data and a simple averaging operation. Compared to the computationally intensive core training loop—which involves multiple epochs of forward and backward propagation (5 local epochs in our experiments)—this step adds less than 0.5% to the local training time. Therefore, the total wall-clock time per round for Fed-ARPL is effectively identical to that of SemiFL and (FL)$^2$.
>
> **Action:** We have added a new Appendix F titled "System Overhead Analysis," which provides the detailed mathematical breakdown of these communication and computation costs to substantiate these claims.

---

> ### Author Response · Authors · 2025-11-28
> **We are looking forward to your feedback.**
>
> Dear Reviewer PQVk,
>
> Thank you again for your time and effort in reviewing our manuscript. We found your comments regarding the robustness of our mechanisms and system overhead to be particularly constructive, and we have done our best to address them through detailed clarifications, quantitative analyses, and manuscript revisions.
>
> Given that the rebuttal period is coming to a close, we would be very grateful if you could let us know whether our responses have sufficiently resolved your concerns, especially regarding the prototype quality safeguards and system costs. We remain fully available to provide any further elaboration or data should you have remaining questions.
>
> Thank you once more for your valuable engagement with our work.
>
> Sincerely,
>
> Authors of Submission 10229

---

### Note · Authors · 2026-01-13

I have read and agree with the venue's withdrawal policy on behalf of myself and my co-authors.